# Rationale and design of a Scale-Up Project Evaluating Responsiveness to Home Exercise And Lifestyle Tele-Health (SUPER-HEALTH) in people with physical/mobility disabilities: a type 1 hybrid design effectiveness trial

James H Rimmer,[1] Tapan Mehta,[2] Jereme Wilroy,[1] Byron Lai,[1] Hui-Ju Young,[1] Yumi Kim,[1] Dorothy Pekmezi,[3] Mohanraj Thirumalai[2]

For numbered affiliations see end of article.

**Correspondence to**
Dr Mohanraj Thirumalai; mohanraj@uab.edu

## ABSTRACT

**Introduction** Rates of physical inactivity among people with physical disabilities are substantially higher than in the general population and access to home-based tailored exercise programmes is almost non-existent. Using a theory-driven eHealth platform, an innovative exercise programme referred to as movement-to-music (M2M) will be delivered as a customised, home-based exercise intervention for adults with mobility disabilities.

**Methods and analysis** Participants are being recruited for this type 1 hybrid design based effectiveness trial through outpatient clinics at a large rehabilitation centre and randomised to one of three groups: (1) M2M, (2) M2M plus social networking (M2M^plus) and (3) attention control (AC). The intervention includes a 12-week adoption phase, 12-week transition phase and 24-week maintenance phase, at which the collection of objective measures on exercise, fitness and self-reported measures on health will be obtained at the start of each phase and at follow-up. The study compares the effectiveness of M2M and M2M^plus in increasing physical activity (primary outcome), adherence, fitness and physical functioning compared with the AC group and examines the mediators and moderators of the treatment effect.

**Ethics and dissemination** The Institutional Review Board of The University of Alabama at Birmingham granted full approval: (IRB-160923002). Dissemination of findings will include publication in peer-reviewed journals, presentations at regional, national and/or international meetings, and the National Center on Health, Physical Activity and Disability (NCHPAD, www.nchpad.org). This study will strengthen our understanding of the potential benefits of eHealth exercise interventions for people with physical disabilities and build on strategies that aim to recruit larger samples in exercise trials.

**Trial registration number** NCT03024320; Pre-results.

## Strengths and limitations of this study

► The trial is the largest of its kind exploring the use of telehealth technology to improve access to regular and sustainable home exercise among an underserved population of adults with mobility disabilities.
► The trial addresses a growing need among physicians treating patients with mobility disabilities to have easily accessible eHealth home exercise videos that can be tailored to the functional level of their patients.
► Using constant monitoring technology will allow researchers to know when participants are not engaged in the intervention and prompt coaching calls to reduce drop-out.
► Recruitment will occur through a single rehabilitation hospital network and may not be generalisable to other regions across the USA and other countries.

function,[1] people with physical disabilities remain one of the least active and obese[2–8] populations in society. In the latest data from the Centers for Disease Control and Prevention, nearly one-half of all adults with disabilities perform no aerobic physical activity.[9] These patterns of low physical activity become more problematic across the lifespan when the effects of the natural ageing process are compounded by years of sedentary living, poor nutrition and severe deconditioning.[10–13]

In addition to the physical impairments associated with a disability which limits physical activity, initiating and maintaining a physically active lifestyle is difficult for people with physical disabilities due to a number of barriers faced in the home, built environment (neighbourhood, community-based exercise

## BACKGROUND

Despite what is known about the positive effects of exercise in improving health and

and recreation facilities) and healthcare system.[14] Dozens of papers have reported that the built environment creates substantial limitations in accessing outdoor and indoor physical activity programmes and venues.[15–18] Outdoor exercise may be challenging or impossible because many neighbourhoods either lack sidewalks or have badly damaged surfaces presenting a high risk of falling among people who have poor balance and/or use wheelchairs.[19] Also, high traffic volume can be intimidating for many people with a mobility disability who have difficulty getting across streets in the time allotted by the traffic light.[20] In terms of getting to an indoor facility such as a fitness centre, lack of accessible transportation is often the number one barrier to using these facilities followed by cost of the membership.[20]

Recent data demonstrate that people with physical disabilities who are advised by their doctor to exercise are 82% more likely to do so.[9] However, the current healthcare system continues to provide little guidance to people with physical disabilities on how to exercise, leaving many, if not most, minimally prepared to manage or improve their health and fitness.[21 22] Therefore, healthcare providers require exercise programmes that they can recommend to their patients with disabilities that are evidence based, easily accessible and tailored to the specific functional and health-related needs of the individual.

In recent years, internet-based interventions targeting health issues such as nutrition, smoking, physical activity or multiple health behaviours have become increasingly popular.[23] These interventions have several advantages for people with physical disabilities. In addition to the interactive and convenient nature of internet-based programmes, health professionals can use web platforms to provide tailored information to individual users and potentially reach large groups of people with physical/ mobility disabilities at relatively low cost. A notable benefit of eHealth programmes for people with physical disabilities includes the ability to participate at home, which is the most convenient place for many people with physical/mobility disabilities to engage in sustained physical activity behaviour. If designed properly, home-based exercise allows for prompt feedback, individual tailoring, and continued personalised guidance and social support.[24] Thus, internet-based exercise programmes create an accessible venue for people with physical disabilities to achieve the exercise regimens prescribed by a health professional and can potentially include individuals who do not have access to conventional community-based exercise programmes.

To fully address the barriers to accessing exercise, including the need for healthcare providers to have off-the-shelf resources that they can quickly provide to their patients, technology can create exercise opportunities in the comfort of a person's home to enhance long-term participation.[25 26] This is an avenue that is currently not available in most communities across the USA. Therefore, the proposed scale-up exercise intervention will enroll a large cohort of adults with mobility disabilities from a university-based outpatient rehabilitation centre, to determine if an innovative technology-based exercise programme at the home can achieve improvements in physical activity adherence and health and functional outcomes. The study is referred to as SUPER-HEALTH, which stands for Scale-Up Project Evaluating Responsiveness to Home Exercise And Lifestyle Tele-Health.

The primary aim of this study is to test the effectiveness of a home-based eHealth exercise programme for increasing physical activity among a clinical population of people with physical/mobility disabilities. The intervention, referred to as movement-to-music (M2M), is being compared with an enhanced version of M2M that includes social networking (M2M$^{plus}$). Both of these interventions are compared with an attention control (AC) group. Secondary aims include those related to intervention effectiveness and implementation. Regarding effectiveness, aims include (1) examining improvements in health (pain, sleep, quality of life) and physical function (balance, strength, endurance) between M2M and M2M$^{plus}$ groups compared with the AC group and (2) assessing potential mediators and moderators of intervention efficacy (eg, social cognitive theory constructs such as self-efficacy, self-regulation, social support, outcome expectancies; and demographic factors eg, age, race, disability type) to understand for whom and how the intervention is effective. Implementation is assessed via participant uptake of the intervention, whereby the aim is to explore participant flow throughout all stages of the study (contact through enrollment and intervention adoption through intervention maintenance).

## METHODS
This paper follows the Standard Protocol Items: Recommendations for Interventional Trials checklist,[27] Template for Intervention Description and Replication checklist,[28] and conduct and reporting of the trial follows Consolidated Standards of Reporting Trials guidelines.[29]

### Study design
SUPER-HEALTH is a single-site, three-arm parallel group type 1 hybrid design effectiveness trial[30 31] evaluating the effects of two intervention groups (M2M and M2M$^{plus}$) compared with AC in people with physical/mobility disabilities, with assessments conducted at four time points: baseline, 12 weeks, 24 weeks and 48 weeks.

### Recruitment
Participants are being recruited from outpatient clinics of a large university-based rehabilitation centre through physicians and their staff and by physical mail-outs from January 2018 until the desired sample size is obtained. Supplementary recruitment strategies include brochures placed in clinics, society events, newsletters, advertisements and word of mouth. Additionally, recruitment will include screening more male participants than females.

In order to enhance the likelihood of enrolment through clinician referral,[32] research staff attend periodic meetings with the physicians in the outpatient centre. Recruitment from outpatient clinics include: (1) direct referrals from physicians and their staff; (2) indirect referrals through advertisement materials in waiting rooms; (3) targeted in-person visits by research staff on the day of a patient's clinic appointment; and (4) routine recruitment visits to the waiting room by research staff. Physicians have been instructed to hand out brochures that include the study objectives and contact information. The identification of eligible participants for in-person visits occurs through the review of medical records and clinic appointments using the password-protected hospital database. After potential participants are identified, the recruitment coordinator greets the individuals in the waiting room, provides an overview of the study objectives and consents individuals interested in participating in the study.

Physical mail-outs are being delivered to potential participants that are identified from the National Institutes of Health (NIH)-funded National Centers for Biomedical Computing based at Partners HealthCare System: Informatics for Integrating Biology and the Bedside (i2b2) (https://www.i2b2.org/about/index.html). The i2b2 has been used to identify study cohorts within the outpatient rehabilitation clinics and address research questions by integrating a wide variety of clinical data sources. After an initial search query of participants located within the same state where the study is being conducted, and who attend one of the outpatient clinics, we identified approximately 5400 people with neuromuscular disorders and musculoskeletal conditions. This list directs the physical mail-outs, which include a flyer that describes the study and a letter cosigned by the chair of department overseeing the clinics and the principal investigator. Mail-outs are being sent in batches of 100 every 2–4 weeks and will be increased or decreased depending on the response rate.

### Exclusion criteria

Individuals with a physical/mobility disability are eligible for the study. In order to remain consistent with other studies,[4] this was defined as self-reported difficulty with one or more of the following activities: (1) walking (some, much, unable to do) without special equipment use; (2) walking one-quarter of a mile; (3) remaining on feet for more than 2 hours; (4) taking 10 uninterrupted steps; (5) kneeling, stooping or crouching; or (5) standing up from an armless straight chair. Individuals who meet the following criteria are not eligible to participate:

► Accumulating more than 60 min of moderate/vigorous physical activity per week.
► Do not report having a diagnosis of a physical/mobility disability.
► Not within working age (18–64 years of age).
► Enrolled in a structured exercise programme over the past 6 months.

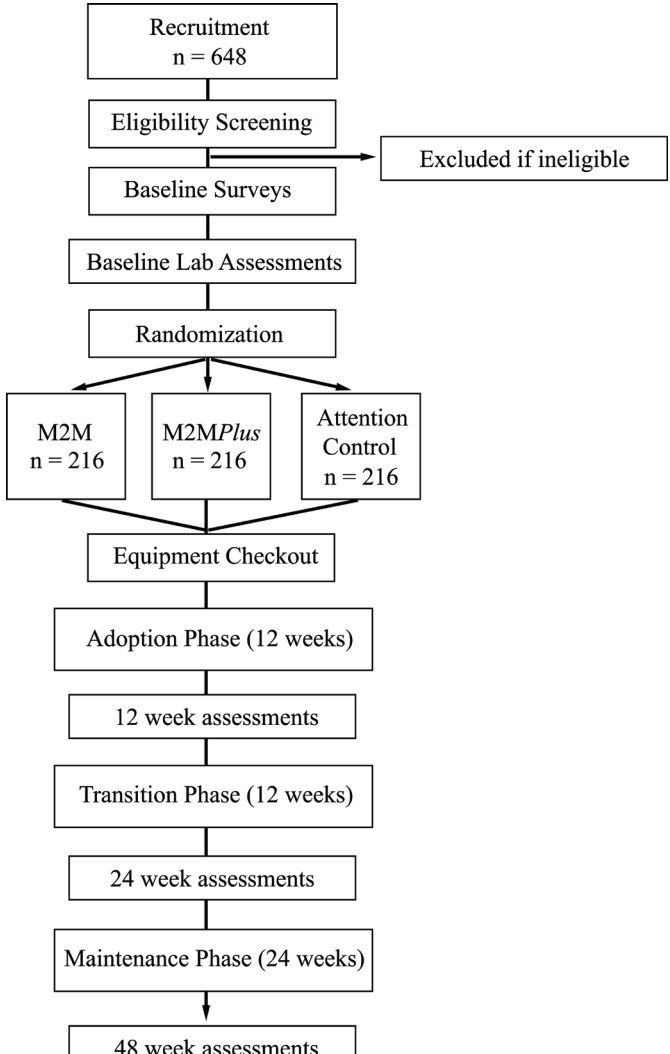

**Figure 1** Study flow chart. M2M, movement-to-music.

► Unable to use upper extremities to exercise.
► Unable to converse and read English.
► Medically unstable to perform home exercise as determined by their physician.
► Cognitive impairment that may preclude self-directed daily activities.
► No Wi-Fi internet access.

### Study procedures

The study flow diagram includes three phases of the intervention, adoption (weeks 1–12), transition (weeks 13–24) and maintenance (weeks 25–48) (figure 1). This phased approach allows for gradual adjustment in the dosage of the intervention and the ability to capture changes within and across phases. Details of the interventions offered in each arm are shown in table 1.

All data storage is established via the Research Electronic Data Capture (REDCap),[33] an electronic data capture system. When potential participants fill in their information via the study website, the data are automatically stored in REDCap. Potential participants' information with other methods of contact (eg, mail-back, call)

**Table 1** SUPER-HEALTH Study design and intervention plan

| Intervention arm | Phase | Health education content | M2M exercise videos | Social networking support |
|---|---|---|---|---|
| Attention control (AC) (Enables estimation of research personnel involvement effect (within group)) | Adoption | Weekly | None | Not offered |
| | Transition | Biweekly | None | Not offered |
| | Maintenance | Monthly | None | Not offered |
| M2M (Enables estimation of home-based M2M (between-group comparison with AC)) | Adoption | Weekly | Weekly new video | Not offered |
| | Transition | Biweekly | Biweekly new video | Not offered |
| | Maintenance | Monthly | Monthly new video | Not offered |
| M2M$^{Plus}$ (Enables estimation of social support of M2M$^{Plus}$ (between-group comparison to M2M)) | Adoption | Weekly | Weekly new video | Offered |
| | Transition | Biweekly | Biweekly new video | Offered |
| | Maintenance | Monthly | Monthly new video | Offered |

M2M, movement-to-music; M2M$^{Plus}$, M2M+social networking; SUPER-HEALTH, Scale-Up Project Evaluating Responsiveness to Home Exercise And Lifestyle Tele-Health.

is manually entered into REDCap by the recruitment coordinator. The recruitment coordinator reaches out to potential participants within 48–72 hours based on their preferred contact methods (ie, phone, emails) for more information regarding the study as well as participation eligibility. If eligible, the participant will be consented over the phone and will then receive baseline surveys to be completed online prior to their scheduled testing visit. Once participants complete the baseline testing, they are randomised.

After group allocation, participants receive a study designated email address that is uploaded to REDCap. Surveys are automatically sent by email at the remaining follow-up time points, 12 weeks, 24 weeks and 48 weeks, to the new email address. There is a 10-day window open before and after the exact date of follow-up data collection based on the initial laboratory testing date. If participants do not complete a survey, a reminder e-mail is automatically sent by REDCap to the participant on a daily basis. A survey that is not completed in 5 days will be followed up by a phone call from research staff.

Fitness and physical function measures are assessed in a research laboratory within a state-of-the-art universally designed community health and fitness facility. Transportation is provided to the participants, if needed. After a review of the consent form, the laboratory staff completes anthropometric measures (height, weight, waist circumference) and vital signs (blood pressure, heart rate). Participant then completes a battery of fitness and physical function measures while laboratory staff record results in a data collection packet.

### Interventions

After the laboratory assessments are completed, the technology coordinator provides the participant with the appropriate equipment based on their group assignment and functional level. The equipment includes a tablet (ASUS Zenpad 3 s 10), a tablet case (Fintie Folio Stand Cover), stylus, Fitbit Charge 2 to monitor physical activity, a set of two pound wrist weights (SPRI Thumb-block Wrist Weight) and an optional stand (Musician's Gear Folding Music Stand). The technology coordinator then familiarises participants with the equipment and also guides participants through the tablet app. Equipment instructions cover basic operation (eg, turning the devices on and off, connecting to Wi-Fi, etc) and maintenance skills (eg, cleaning the devices and damage prevention). Regarding app use, participants are carefully guided through the tablet app content and functions. Following these instructions, participants are then asked to independently navigate through the app and view all content.

The tablet app includes the foundational elements required for rapid deployment of the video and textual content, and is password protected to enable detailed usage tracking. The home page of the app provides brief weekly objectives and links to newly received content. Other features of the app include a video and article library, calendar for scheduling and activity dashboard with Fitbit data. In addition to these features, the M2M$^{plus}$ group's version of the app includes elements that support social interaction. These ancillary features include the ability to private message other users, leaderboards that display user progress, a user profile, newsfeed and the ability to comment on and 'like' the exercise videos. The app version for the intervention groups (M2M and M2M$^{plus}$) includes exercise video content, while the AC group will only receive access to the infographics and the Fitbit dashboard.

### Educational resources

Each group receives textual content (ie, infographics articles) aimed at improving health through lifestyle changes and tailored to adults with physical/mobility disabilities. This content includes physical activity recommendations, how to maintain activity, developing healthy nutrition habits, ways to reduce stress and other health-related information. Each group receives this content weekly for 12 weeks (adoption phase), biweekly for the following 12 weeks (transition phase) and monthly for the last 24 weeks (maintenance phase).

## Exercise video content

Exercise videos include movements that were adapted from an onsite programme that has been conducted at a state-of-the-art fitness facility for people with disabilities. The onsite programme incorporated an extensive set of movement routines that were choreographed to the functional needs of people with a range of physical disabilities. For the present study, we modified the onsite programme into a video-based package that could be performed independently in the home. Programme modifications were made using a framework referred to as Modality, Adaptation, Position, Pattern, Equipment, Tempo and Time (MAPPETT) to systematically document exercise adaptations. The Modality (range of motion/flexibility, muscle strength/endurance, cardiorespiratory endurance and balance), Tempo (a given range of music beat per minute) and Time (duration of each exercise routine) are intervention elements that are structured and need to be delivered accordingly. The Position (ie, standing or sitting), Pattern (ie, anatomical planes, types of motion, static/dynamic, movement sequence) and Equipment (ie, wrist weight, ankle weight, chair) are elements that can be adapted (Adaptation) to people with different functional capacities. The goal of the home-based programme is to slowly and safely progress participants towards achieving and maintaining 150 min of moderate-intensity physical activity per week, which is based on the guidelines provided by the American College of Sports Medicine.[34 35]

The exercise intervention is referred to as M2M. The M2M intervention includes three broad subprogrammes or levels that can be arranged by programme staff to meet the functional needs of the participant: (1) able to use all four limbs; (2) able to use only the upper limbs (eg, someone with lower extremity paralysis) and (3) able to use only one side of the body to exercise (eg, individuals with stroke/hemiparesis). The determination of the level for each participant is based on their baseline physical function assessments. Each M2M session includes a set of videos that can include up to four types of exercise routines: flexibility, muscle strength/endurance, cardiorespiratory endurance and balance. Each session ends with a cool down breathing routine. Participants are instructed to complete the assigned M2M exercise videos three times per week. The session duration and exercise intensity are gradually increased throughout the programme.

The exercise videos are delivered across the three intervention phases: (1) adoption, weeks 1–12; (2) transition, weeks 13–24 and (3) maintenance, weeks 25–48. During the adoption phase, a new set of videos are uploaded each week to the eHealth platform. The exercise programme begins with a total of 15 min of exercise at week 1 and slowly progresses to 90 min at week 12. The rationale for starting the programme at a low duration is to help participants build confidence and gradually incorporate the exercise programme into their daily life. Each week the videos include newly added routines that are guided by an M2M instructor and an actor with a disability that matches the functional level of the participants. When a new routine is introduced, the first segment of the video shows the M2M instructor breaking down the routine and explaining each movement pattern (guided exercise portion). In the following week, the guided exercise portion of the same routine is removed from the video so that the participants can perform the routine directly with music. Participants have the ability to view previously completed exercise videos in a video library within the tablet app.

During the transition and maintenance phases, one new video is added biweekly (twice a month) and monthly, respectively. The goal of the transition phase is to progress participants to a goal of 150 min of moderate-intensity exercise.[34 35] The transition phase starts with three 30 min sessions at week 13 of the intervention (week 1 of this phase) and ends with three 50 min sessions at week 24 (150 min/week). For the maintenance phase, the session duration remains the same and there are new monthly videos added as an incentive to encourage participants to adhere to the study protocol by using new M2M routines.

## Social networking system

A social networking system (SNS) is integrated into the eHealth platform for the M2M$^{plus}$ group. The SNS contains the essential building blocks of social media including (1) identity, (2) sharing, (3) presence, (4) relationships, (5) reputations, (6) groups and (7) conversations.[36] Additionally, M2M$^{plus}$ participants join a weekly video conference group discussion session led by the project coordinator who has expertise in motivational interviewing strategies. The purposes of these sessions are to stimulate discussion among participants and to encourage them to support each other. Each session incorporates behavioural change strategies that have been modified from previous research with adults with physical disabilities.[37] The sessions focus on topics related to constructs from the social cognitive theory (eg, social support, overcoming barriers) and preventing relapse.

## Theoretical framework

The intervention is grounded in the social cognitive theory,[38] which provides a comprehensive, multilevel model of behavioural change emphasising personal and environmental factors that lead to the adoption and/or maintenance of health-promoting behaviour.[39] According to the social cognitive theory, health behavioural change is based on reciprocal determinism among three domains: (1) the behaviour, which involves performance or mastery by the individual (behavioural capability, self-regulation); (2) personal factors that involve cognitive, affective and biological events (eg, self-efficacy) and (3) the environmental factors that facilitate or impede change (social and physical environmental factors).[39] Thus, the intervention targets exercise self-efficacy through self-regulation strategies, incremental goal setting, reinforcements using infographics (all groups) and by including adults with physical/mobility disabilities

as demonstrators for exercise videos (M2M and M2M$^{plus}$ only). These approaches focus on mastery experiences, social modelling and verbal persuasion and thus are well aligned with Bandura's research on the key influences on self-efficacy.[40] Social support for physical activity is targetted using social networking (eg, messages, posts, newsfeeds)[41 42] and group discussion sessions[43] led by research staff (M2M$^{plus}$ only).

### AC group
In addition to the Fitbit, the AC group receives a tablet that only includes the educational materials. After the 48-week assessment is completed, participants receive access to the exercise videos.

### Randomisation
Eligible participants are randomised into one of three arms with 1:1:1 allocation ratio using a permuted block randomisation approach where the block size is unknown to the intervention staff. The randomisation sequence is generated a priori using a computer-generated random schedule in a permuted block (SAS V.9.4). The randomisation schedule is then embedded into a randomisation module in REDCap.[33] This system allows researchers to

manage the information with a higher level of security, remove physical envelop and set an individual level of blinding within the system.

Staff performing recruitment, outcome measurements and data entry for the primary outcomes are blinded to group allocation. However, due to the nature of the intervention, it is not possible to blind the staff administering the interventions or the participants. Participants are instructed not to inform the data collection staff of their intervention status when they return for follow-up measures.

### Intervention fidelity
The intervention uses the latest eHealth technology as a means to deliver and monitor the intervention. Each participant receives the latest in activity monitoring (Fitbit), streaming, content (M2M and M2M$^{plus}$ only) and a social networking platform (M2M$^{plus}$ only), which paired with the use of cloud-based technology allows for the intervention to be administered in the home. Our intervention fidelity plan (table 2) is based on the best practices and recommendations from the NIH Behaviour Change Consortium (BCC). The BCC recommendations

| Table 2 | Intervention fidelity plan |
|---|---|
| **Treatment fidelity element** | **Strategies** |
| Study design | Ensure intervention consistency with social cognitive theory. |
| | Protocol for eHealth platform maintenance and resolution of technical problems during study. |
| | Weekly delivery and notification of movement-to-music (M2M) videos to motivate participants to use. |
| | Phone coaching would assess the use of the eHealth platform and motivate intervention participants to complete their sessions. |
| | Password protection to ensure that participants access only material belonging to their study arm. |
| | Wearing of Fitbit will be monitored and if no data shows for 7 days, the participant will be contacted by phone and/or email. |
| | Intervention delivered via eHealth platform, not via clinicians. |
| Provider training | Training of all individuals who will be conducting the screening, phone coaching, and interviews. Manual of operating procedures to contain standardised scripts for all interactions. Booster training sessions conducted every 6 months. |
| | Intervention will be delivered through eHealth platform in exactly the same way to all participants of the two intervention arms. |
| Delivery of intervention | Prescribed behaviours (aerobic/strength moves), as defined by the M2M protocol, performed by the interventionist in M2M videos will be qualitatively and quantitatively assessed. |
| | Technology acceptance will be measured through perceived usefulness and ease of use scale that will provide information about the usability and the extent to which the participant has positive perceptions and intentions to use the platform. |
| | The research coordinator will regularly monitor the eHealth platform throughout the study to ensure videos and other materials were working correctly. |
| | Technical difficulties with the delivery of the intervention will be resolved in timely manner. |
| | The number of participant logins, clicks to video links and minutes watched will be monitored by tools on the intervention eHealth platform server. |
| Receipt of intervention | Every time a participant views a M2M video, a timestamp and minutes viewed are recorded. |
| | Each time the participant reads an article or opens a video, they have a check in button they press. |

**Table 3** Measures, instruments and time points

| Variables | Instruments | Time point | Role |
|---|---|---|---|
| Physical activity[1] | Godin Leisure-Time Exercise Questionnaire[41] | B, A, T, M | Primary outcome |
| Cardiorespiratory endurance[1] | Submaximal Arm Ergometer Spirometry | B, A, M | Secondary outcome |
| Strength[1] | Grip strength | B, A, M | Secondary outcome |
| Adherence[1] | Tracking data | ongoing | Secondary outcome |
| Secondary health conditions[2] | National Institutes of Health Patient-Reported Outcome Measurement Information System questionnaires (pain, fatigue, sleep, quality of life) | B, A, T, M | Secondary outcome |
| Social cognitive theory constructs[2] | Exercise Self-Efficacy Scale[53] Exercise Goal-setting Scale[54] Multidimensional Outcomes Expectations for Exercise Scale[55] Barriers in Physical Activity Questionnaire-Mobility Impairment[56] Social Provisions Scale[57] | B, A, T, M | Mediation |
| Demographics[2] | Questionnaire | Baseline | Covariate |
| Health history[2] | Questionnaire | B, A, M | Covariate |
| Physical function[1] | Timed up and go, repeated chair stands, Timed 25-foot walk | B, A, M | Secondary outcome |
| Blood pressure[1] | Blood pressure monitor | B, A, M | Covariate |
| Anthropometrics[1] | Height, weight, BMI, waist circumference | B, A, M | Covariate |
| App quality and usability | eHealth Literacy Scale[65] Systems Usability Scale[66], Mobile Application Rating Scale[67] | B, M A | Covariate |

are intended to link theory and application across five primary study phases, including study design, provider training, monitoring and improving the delivery of the intervention, and monitoring and improving the enactment of intervention skills.

### Outcome measures

Laboratory-based research outcome measures are administered at baseline, 12 weeks and 48 weeks by independent evaluators blind to treatment assignment. Survey measures are automatically delivered by the REDCap system at baseline, 12 weeks, 24 weeks and 48 weeks.

The primary outcome is changes in rates of exercise participation, measured via self-report by the Godin Leisure-Time Exercise Questionnaire (GLTEQ),[44] at the four time points. Secondary effectiveness outcomes include health and physical function measures, which involve a battery of tests performed at baseline, 12 weeks and 48 weeks (see table 3). In addition, anthropometric measures and vital signs are assessed at each testing visit. Short forms developed by the NIH patient-reported outcome measurement information system (PROMIS) are used to measure symptoms and quality of life indicators. The PROMIS short forms used for this study include sleep disturbance, pain, depression, anxiety, physical function and social interactions. Social cognitive theory measures include surveys and scales on exercise goal setting, barriers to physical activity, outcome expectations for exercise, social support and exercise self-efficacy. Other measures include app usability, eHealth literacy assessment and demographics (ie, race, sex, age). A brief tabulation of the measures for this study including information on key outcome variables, covariates and time points is shown in table 3.

Outcomes related to participant uptake include quantitative measures of participant flow throughout the intervention. Data for participant flow include the number of individuals recruited, screened and enrolled, along with intervention adherence (weeks 1–24) and maintenance data (weeks 25–48). Within these stages, the number of participants that withdraw and the reasons for withdrawal are recorded. Adherence is defined as the percentage the participant meets or exceeds the study weekly exercise protocol, which is measured by how many minutes the individual views the exercise videos divided by the weekly prescription. The week 1 video begins with 5 min of exercise, and each participant is asked to complete the videos three times each week equalling a total of 15 min for the first week. This increases approximately 5 min each week until reaching 150 min of exercise video viewing per week (50 min of exercise completed three times) by the end of the transition phase (week 24). Maintenance is defined as the percentage the participant continues to meet the weekly exercise protocol prescribed from weeks 25–48, and the protocol will remain at 150 min per week.

To improve adherence and ensure delivery of the intervention, the project coordinator monitors weekly data uploaded from the Fitbit and tablet that include the date, total steps per day, and a log for the exercise videos, which includes a timestamp, title of video and the number of minutes spent viewing. If there is no Fitbit data for more than 7 days or if the amount of video minutes completed in a given week is below 25% of those prescribed, the participant is contacted by research staff trained in motivational interviewing techniques. The purpose of the call is to gather information on non-adherence and aid the participant in creating an action plan for meeting the prescribed activity. Adherence across groups will be compared via: (1) the percentage of days that individuals use the Fitbit compared with those prescribed (number of times worn divided by total sessions) and (2) attrition (number of individuals that withdraw from the trial).

### Power and sample size

A sample size of 648 participants resulted from conducting conservative sample size calculations with minimal assumptions, which involved the following assumptions: 80% power, two-sided t-test comparing changes in the primary outcome of physical activity, familywise error rate of 0.05 to account for multiple testing, attrition rate (AR) of 36% and intention-to-treat analyses. Each pairwise comparison for the primary outcome (M2M vs AC, M2M$^{plus}$ vs AC and M2M vs M2M$^{plus}$) is considered as a family of hypotheses. We did not account for multiple comparisons (between different arms) since these are planned a priori. A sample size of 648, which results in 415 participants as completers after accounting for the 36% attrition, provides 90% power to detect an effect size (ES) of 0.32 (Cohen's d) between any of the pairwise comparisons aforementioned above.

We can consider the two sets of secondary outcomes (four objective measures and four patient reported-measures—see table 3) as separate families of hypotheses. Again, we do not account for multiple comparisons (pairwise comparison between three arms) but do account for multiple outcomes testing within each pairwise comparison. Hence, assuming a type 1 error rate of 0.0125 with a conservative Bonferroni correction for multiple testing, we will have 90% power to detect an ES of 0.373.

Previous ESs estimated for changes in exercise from two web-based interventions reported ES of 0.6129 and 0.8, respectively,[45 46] and ES achieved from two home-based behavioural interventions to increase exercise in wheelchair users above 0.5.[47 48] Our minimum detectable ES of 0.375 is much smaller than the aforementioned ESs to account for the variability in the ESs and winner's curse[49] (anticipated scenario given that we aim to recruit a far larger sample size). An assumed AR of 36% was based on the study by Froehlich-Grobe *et al*.[48] While our prior studies had an AR as low as 13%, the eHealth literature cautions of higher ARs. Achieving lower ARs may possibly relate to the role of health/motivational coaches in the study. Finally, we will be able to bring more efficiency

in terms of power and ES since our primary statistical analysis is based on a mixed modelling approach. The mixed modelling approach accounts for the correlation between measurements from the same participant that improves power for the same ES or provides us the ability to detect a lower ES for the same power and aforementioned assumptions.[49] Mixed modelling technique also uses all the data available (ie, imputes any missing outcome data). Hence, despite attrition, we will have a larger sample size while conducting longitudinal analyses, which would again bring more power/sample efficiency.

### Analyses

All statistical analyses are conducted in an intention-to-treat manner, at the individual level. For the single primary outcome, statistical significance will be evaluated at a two-tailed hypothesiswise error rate of 0.05. Consistent with published guidelines for statistical reporting, exact p values (rather than, eg, 'p<0.05' or 'NS') will be reported.[50] This allows readers who may have their own opinions about multiple comparison corrections to know the raw uncorrected result and make their own judgements about statistical significance especially given that the significance tests conducted for the primary outcome are specified a priori and modest in number. As Saville[51 52] noted, in multiple comparison issues, there is no right answer and each investigator must ultimately 'cut the Gordian knot' themselves. For the secondary outcomes, statistical significance is evaluated at a familywise error rate of 0.05 after accounting for multiple outcomes testing. Based on the p values for all analyses reported, we will also report the false discovery rate and false discovery proportion.

In general, missingness in outcomes is handled by mixed models for repeated measures data. Quality control includes descriptive and graphical approaches to summarise baseline characteristics of all key variables.

For the primary outcome of the physical activity, we will compare two planned pairwise comparisons. The primary aim includes two hypotheses stating that the M2M and M2M$^{plus}$ interventions will lead to greater increases in the exercise from baseline to the transition phase when compared with the AC group. The outcome variable will be a time-varying measure (two time points: baseline and transition phase) of exercise. Using self-report data from the GLTEQ, a Health Contribution Score will be calculated.[44] The baseline exercise measure will be calculated as the average exercise week across a 7-day period prior to starting the intervention (adoption phase). Similarly, the average exercise for a 7-day period at the end of each intervention phase will be estimated. The main hypothesis addresses the change in activity between baseline and transition phases across intervention groups. Baseline covariate measures will be included if differences are identified across intervention arms, or if the inclusion of these covariates improves the corrected Akaike information criterion.[53 54]

In addition to any baseline covariates, fixed effects will include intervention arms (M2M, M2M$^{plus}$), AC, time (baseline, transition) and an arms X time interaction. Contrast statements will be used to test the null hypotheses in conjunction with the fitted model coefficients. The estimates of the change in exercise for each arm and their 95% CI will also be reported. Secondary data analyses will include testing for non-linearity in the differences between change scores in exercise across the intervention phases and testing whether there were greater increases in exercise in M2M and/or M2M$^{plus}$ compared with AC from baseline to end of adoption and baseline to transition.

### Ethics and dissemination

The institutional review board will remain informed of any protocol changes or adverse events for the safety of participants. The recruitment and project coordinators will consent each participant to the study. Once the participant signs the informed consent, they receive a unique research ID code and no personally identifiable information will be linked to data collected for this study. Quality assurance procedures are used to minimise missing data, possible errors and to correct errors before the final database lock. After completion of data entry, two additional research staff (project coordinator and data reviewer) audit data periodically and lock the record. When the survey is submitted to REDCap, research staff receive a notification via email and review the data within 48 hours. An independent data safety and monitoring committee is not necessary due to the minimal risk associated with participant outcomes. All data collection will be overseen by the principal investigator and coinvestigators, including a biostatistician and physician.

Findings from this study will be shared publicly and disseminated by: (1) publication in peer-reviewed journals and/or (2) presentations at regional, national and/or international meetings. Findings will also be disseminated through the National Center on Health, Physical Activity and Disability (NCHPAD, www.nchpad.org), which has over 20 000 email subscribers. If the M2M exercise intervention achieves successful outcomes, we will make the programme available to other medical centres through the NCHPAD website. We are committed to the sharing of final research data, being mindful that the rights and privacy of our research participants must be protected at all times, that there is the need to protect patentable and other proprietary data (ie, our web-based platform), and that restrictions on data sharing may be imposed by agreements with third parties. Published data with non-identifiers will also be shared with the 2018 Physical Activity Guidelines committee on request.

### Patient and public involvement

There was no patient or public involvement in the design of this study.

## DISCUSSION

This paper has presented the background and design for a randomised controlled trial investigating the effectiveness of a home-based, eHealth exercise programme for adults with physical/mobility disabilities. To our knowledge, this study is the largest exercise trial ever conducted on people with physical/mobility disabilities.

Engaging in the recommended levels of physical activity (150 min of moderate-intensity physical activity per week) can be extremely challenging for many people with disabilities, who have limited to no access to home exercise equipment, and indoor and outdoor physical activity programmes and venues (eg, outdoor parks, fitness facilities).[15–18 55] In particular, outdoor exercise, used by many people who do not have a mobility disability (eg, walking, jogging, hiking, cycling), may be difficult or impossible to perform by some people with physical/mobility disabilities. This is because neighbourhoods either lack sidewalks or have surfaces that are badly damaged imposing a risk of falls; high traffic volume makes it cumbersome to get across streets; hilly terrain may be too difficult to traverse and for people who use wheelchairs, manually pushing a chair for an extended period of time to achieve the recommended levels of exercise may induce or worsen shoulder pain.[56] Indoor exercise at a local fitness centre has also been documented as being largely inaccessible to people with physical/mobility disabilities due to lack of accessible exercise equipment and programmes and[57 58] transportation. Membership costs often exceed economic resources.[55]

The low levels of exercise participation observed in people with physical/mobility disabilities likely have a negative impact on their health. Nearly one-half of people with disabilities are physically inactive, which the US Centers for Disease Control and Prevention defines as not engaging in any aerobic physical activity, and this large inactive subgroup is 50% more likely to have a chronic disease compared with those who get the recommended amount of aerobic physical activity on a weekly basis.[9] Unfortunately, the current US healthcare system is woefully underprepared to prescribe recommended exercise to people with mobility disabilities, leaving many, if not most, minimally prepared to manage or improve their own health and fitness as they grow older and have more health issues.[21 22 59] Healthcare providers clearly need more exercise resources that they can recommend to their patients with disabilities that are evidence based, easily accessible and tailored to their specific functional level.

The SUPER-HEALTH project addresses this growing epidemic of physical inactivity among people with physical/mobility disabilities by providing simple access to an exercise programme tailored to the individuals' functional level and performed in the comfort of their home. Delivery of the exercise programme through a tablet app offers opportunities to (1) reach isolated populations who have no other way to obtain regular exercise tailored to their unique needs; (2) eliminate transportation

difficulties needing to get to a facility-based intervention; and (3) allow the individual to exercise at a time of the day that is convenient for them. Other benefits include their low cost (compared with onsite programmes) and the ability for individuals to work at a self-selected pace in a comfortable environment.[24 60]

There are several novel features embedded into SUPER-HEALTH. First, eHealth (telehealth) technology's capacity to provide patient engagement related to health/wellness can create a connected healthcare model that links individuals to trained health/exercise professionals and their healthcare provider offering them greater participation in their own care. Second, with transportation and programme costs being two of the most common barriers reported among people with mobility disabilities in terms of obtaining regular exercise,[17 61–64] home-based programmes with cloud-based technology and social support hold strong potential for reaching them in their home. Third, the telehealth technology used in this study is complemented by the latest innovation in activity monitoring. The Fitbit is a state-of-the-art accelerometer that provides an objective measure of physical activity allowing the participant to receive activity monitoring throughout the programme. Fourth, the internet-based home training package (aerobics, strength, flexibility and balance) is provided in the form of music, which offers greater levels of enjoyment and increases its potential for future scalability. The exercise routines can be mixed and matched with new music and movement patterns, which keep the exercise routines novel. Finally, the use of an eHealth platform has the potential to shift relatively large segments of the population of people with physical/mobility disabilities from inactive to physically active in a sustainable way that extends beyond the intervention time frame.

## CONCLUSION

SUPER-HEALTH is a multilevel scale-up exercise eHealth intervention using healthcare providers as the point of entry for enrolling patients with physical/mobility disabilities into the study. If SUPER-HEALTH proves efficacious, promotion of health-enhancing exercise for people with physical/mobility disabilities through an eHealth M2M intervention (targeted to individual functional level) has strong potential to reach large numbers of individuals who are currently not being prescribed a home-based exercise programme by their healthcare provider.

**Author affiliations**
¹UAB/Lakeshore Research Collaborative, University of Alabama at Birmingham, Birmingham, Alabama, USA
²Department of Health Services Administration, University of Alabama at Birmingham, Birmingham, Alabama, USA
³Department of Health Behavior, University of Alabama at Birmingham, Birmingham, Alabama, USA

**Contributors** JHR provided content for the introduction and discussion. TM provided content for the data analysis and power calculation sections. MT provided all content on app development and social networking section. JW compiled the first full draft of the manuscript, providing content for recruitment section and made all revisions provided by coauthors. BL provided all content on equipment and technology. H-JY provided all content for the exercise video section. YK provided content on data management protocol. DP provided content for theoretical framework section. All authors reviewed paper in full and provided revisions on the entire paper.

**Funding** This work is supported by a grant from the National Institutes of Health, Eunice Kennedy Shriver National Institute of Child Health and Human Development (1R01HD085186). This study is sponsored by the University of Alabama at Birmingham and will be coordinated by the Lakeshore Foundation, Birmingham, Alabama.

**Disclaimer** The study funder will have no role in the conduct or evaluation of the trial, nor the authority over the study design, analysis, interpretation of results or preparation of manuscripts or abstracts.

**Competing interests** None declared.

**Patient consent for publication** Not required.

**Ethics approval** This protocol was approved by the University's Institutional Review Board (IRB-160923002).

**Provenance and peer review** Not commissioned; externally peer reviewed.

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
