## [Reviewer comments · BMJ Open]

BMJ Open

BMJ Open is committed to open peer review. As part of this commitment we make the peer review history of every article we publish publicly available.

When an article is published we post the peer reviewers' comments and the authors' responses online. We also post the versions of the paper that were used during peer review. These are the versions that the peer review comments apply to.

The versions of the paper that follow are the versions that were submitted during the peer review process. They are not the versions of record or the final published versions. They should not be cited or distributed as the published version of this manuscript.

BMJ Open is an open access journal and the full, final, typeset and author-corrected version of record of the manuscript is available on our site with no access controls, subscription charges or pay-per-view fees (<http://bmjopen.bmj.com>).

If you have any questions on BMJ Open's open peer review process please email info.bmjopen@bmj.com

BMJ Open

Rationale and Design of a Scale-Up Project Evaluating Responsiveness to Home Exercise and Lifestyle Tele-Health (SUPER-HEALTH) in People with Mobility Disabilities

Journal:	BMJ Open
Manuscript ID	bmjopen-2018-023538
Article Type:	Protocol
Date Submitted by the Author:	17-Apr-2018
Complete List of Authors:	Rimmer, James H. ; University of Alabama at Birmingham, UAB/Lakeshore Research Collaborative Mehta, Tapan; University of Alabama at Birmingham, Department of Health Services Administration Wilroy, Jereme; University of Alabama at Birmingham, UAB/Lakeshore Research Collaborative Lai, Byron; University of Alabama at Birmingham, UAB/Lakeshore Research Collaborative Young, Hui-Ju; University of Alabama at Birmingham, UAB/Lakeshore Research Collaborative Kim, Yumi; University of Alabama at Birmingham, UAB/Lakeshore Research Collaborative Pekmezi, D; University of Alabama at Birmingham, Department of Health Behavior Thirumalai, Mohanraj; University of Alabama at Birmingham, Department of Health Services Administration
Keywords:	telerehabilitation, mobility disabilities, telehealth, exercise, disability, mhealth

**Rationale and design of a Scale-Up Project Evaluating Responsiveness to Home Exercise**
**and Lifestyle Tele-Health (SUPER-HEALTH) in people with mobility disabilities**

James Rimmer,¹ PhD, Tapan Mehta,¹ PhD, Jereme Wilroy,¹ PhD, Byron Lai,¹ PhD, Hui-Ju
Young,¹ PhD, Yumi Kim,¹ MA, Dorothy Pekmezi,¹ PhD, Mohanraj Thirumalai,¹ PhD

**Author affiliations**

¹University of Alabama at Birmingham, Birmingham, USA

**Author Contributions**

JR provided content for the introduction and discussion. TM provided content for the data
analysis and power calculation sections. MT provided all content on app development and
social networking section. JW compiled the first full draft of the manuscript, providing content for
recruitment section, and made all revisions provided by coauthors. BL provided all content on
equipment and technology. HY provided all content for the exercise video section. YK provided
content on data management protocol. DP provided content for theoretical framework section.
All authors reviewed paper in full and provided revisions on the entire paper.

**Keywords:** eHealth, Physical Activity, Physically Disabled, mHealth, Telehealth

**Word count:** 5,384

**Corresponding Author:**

Mohanraj Thirumalai

1720 2nd Avenue South

Birmingham, AL 35294-0113

mohanraj@uab.edu

205-934-7189

**Word Count:** 5385

**ABSTRACT**

**Introduction:** Rates of physical inactivity among people with physical disabilities are
substantially higher than in the general population while access to home-based exercise
programs tailored are almost non-existent. Using a theory-driven eHealth platform, an

innovative exercise program referred to as movement-to-music (M2M) will be delivered as a
customized, home-based exercise intervention for adults with mobility disabilities.

**Methods and analysis:** Participants are being recruited for this randomized controlled trial
through outpatient clinics at a large rehabilitation center and randomized to one of three groups:
a) M2M; b) M2M plus social networking (M2M^{plus}); and c) attention control (AC). The intervention
includes a 12-week adoption phase, 12-week transition phase, and 24-week maintenance
phase, at which the collection of objective measures on exercise, fitness and self-reported
measures on health will be obtained at the start of each phase and at follow-up. The study
compares the effectiveness of M2M and M2M^{plus} in increasing physical activity, adherence,
fitness, and physical functioning compared to the AC group and examine the mediators and
moderators of the treatment effect.

[revised manuscript text omitted]

To fully address the barriers to accessing exercise, including the need for healthcare
providers to have off-the-shelf resources that they can quickly provide to their patients,
technology can play a substantial role in the delivery of home-based exercise to people with
physical disabilities. Providing exercise opportunities in the comfort of a person's home would
increase the opportunity for regular and sustainable exercise that is currently not available in
most communities across the U.S. The proposed scale up exercise intervention will enroll a
large cohort of adults with mobility disabilities from a university-based outpatient rehabilitation
center to determine if an innovative home-based exercise program can achieve improvements
in physical activity adherence, health and functional outcomes. The study is referred to as

SUPER-HEALTH, which stands for *Scale-Up Project Evaluating Responsiveness to Home*
*Exercise And Lifestyle Tele-Health*.

This study aims to test the effectiveness of a home-based e-Health exercise program for
increasing physical activity among a clinical population of people with mobility disabilities. The
intervention, referred to as Movement-2-Music (M2M), is being compared to an enhanced
version of M2M that includes social networking (M2M^{plus}). Both of these interventions are
compared to an attention control (AC) group. Secondary aims include (1) estimating
improvements in health (pain, sleep, quality of life) and physical function (balance, strength,
endurance) between M2M and M2M^{plus} groups compared to the AC group; and (2) assessing
mediators and moderators of social cognitive theory (self-efficacy, social support, outcome
expectancies) and demographic factors (age, race, disability type) of the hypothesized
treatment effect to understand for whom and how the intervention is effective.

**METHODS**

This protocol was approved by the University's Institutional Review Board (IRB-
160923002) and is registered with ClinicalTrials.gov (#NCT03024320) as a phase III clinical
trial. This paper follows the SPIRIT checklist²⁵ and conduct and reporting of the trial follows
CONSORT guidelines.²⁶

**Study design**

SUPER-HEALTH is a single site, three-arm parallel group randomized controlled trial
evaluating the effects of two intervention groups (M2M and M2M^{plus}) compared to AC in people
with mobility disabilities, with assessments conducted at four time points: baseline, 12 weeks,
24 weeks, and 48 weeks.

**Recruitment**

Participants are being recruited from outpatient clinics of a large university-based
rehabilitation center through physicians and their staff and by physical mail-outs from January
2018 until the desired sample size is obtained. Supplementary recruitment strategies include

brochures placed in clinics, society events, newsletters, advertisements, and word of mouth.

Additionally, recruitment will include screening more male participants than females.

In order to enhance the likelihood of enrollment through clinician referral,²⁷ research staff
attend periodic meetings with the physicians in the outpatient center. Recruitment from
outpatient clinics include: 1) direct referrals from physicians and their staff; 2) indirect referrals
through advertisement materials in waiting room; 3) targeted in-person visits by research staff
on the day of a patient's clinic appointment; and 4) routine recruitment visits to the waiting room
by research staff. Physicians have been instructed to hand out brochures that include the study
objectives and contact information. The identification of eligible participants for in-person visits
occurs through the review of medical records and clinic appointments using the password-
protected hospital database. After potential participants are identified, the recruitment
coordinator greets the individuals in the waiting room, provides an overview of the study
objectives, and consents individuals interested in participating in the study.

Physical mail-outs are being delivered to potential participants that are identified from
the NIH-funded National Center for Biomedical Computing based at Partners HealthCare
System: Informatics for Integrating Biology and the Bedside (i2b2)
[<https://www.i2b2.org/about/index.html>]. The i2b2 has been used to identify study cohorts within
the outpatient rehabilitation clinics and address research questions by integrating a wide variety
of clinical data sources. After an initial search query of participants located within the same state
where the study is being conducted, and who attend one of the outpatient clinics, we identified
approximately 5,400 people with neuromuscular disorders and musculoskeletal conditions. This
list directs the physical mail-outs, which include a flyer that describes the study and a letter co-
signed by the chair of department overseeing the clinics and the principal investigator. Mail-outs
are being sent in batches of 100 every 2 to 4 weeks and will be increased or decreased
depending on the response rate.

Exclusion criteria

Individuals with a mobility disability are eligible for the study. In order to remain consistent with other studies,⁴ this was defined as self-reported difficulty (1) walking (some, much, unable to do) without special equipment use; (2) walking one-quarter of a mile; (3) remaining on feet for more than two hours; (4) taking 10 uninterrupted steps; (5) kneeling, stooping, or crouching; or (5) standing up from an armless straight chair. Individuals who meet the following criteria are not eligible to participate:

- accumulating more than 60 minutes of moderate/vigorous physical activity per week;
- do not report having a diagnosis of a physical/mobility disability;
- not within working age (18 to 64 yrs. of age);
- currently enrolled in a structured exercise program over the past 6 months;
- unable to use upper, lower or both sets of extremities to exercise;
- unable to converse and read English;
- medically unstable to perform home exercise as determined by their physician;
- cognitive impairment that may preclude self-directed daily activities
- no Internet access.

Study Procedures

The study flow diagram includes three phases of the intervention, Adoption (weeks 1 to 12), Transition (weeks 13 to 24), and Maintenance (weeks 25 to 48) (Figure 1). This phased approach allows for gradual adjustment in the dosage of the intervention and the ability to capture changes within and across phases. Details of the interventions offered in each arm are shown in Table 1.

[Insert Figure 1]

[Insert Table 1]

All data storage is established via the Research Electronic Data Capture (REDCap),²⁸ an
electronic data capture system. When potential participants fill in their information via the study
website, it is automatically stored in REDCap. Potential participants' information with other
methods of contact (e.g., mail-back, call) is manually entered into REDCap by the recruitment
coordinator. The recruitment coordinator reaches out to potential participants within 48-72 hours
based on their preferred contact methods (i.e., phone, emails) for more information regarding
the study as well as participation eligibility. If eligible, the participant will be consented over the
phone and will then receive baseline surveys to be completed online prior to testing visit. Once
participants complete the baseline testing they will be randomized.

After the group allocation, participants receive a study designated email address which
is uploaded to REDCap. Surveys are automatically sent at the remaining follow-up time points,
12 weeks, 24 weeks, and 48 weeks, to the new email address. There is a 10-day window open
before and after the exact date of follow-up data collection based on the initial lab testing date.
Participants receive the same email a maximum of five times (1 per day) if they do not complete
the survey. When a research staff member notices the missing data after 5 days, the participant
is contacted by phone.

Fitness and physical function measures are assessed in a research lab within a state-of-the-art universally designed community health and fitness facility. Transportation is provided to the participants, if needed. After a review of the consent form, the lab staff completes anthropometric measures (height, weight, waist circumference) and vital signs (blood pressure, heart rate). Participant then complete a battery of fitness and physical function measures (see Table 2) while lab staff record results in a data collection packet.

Interventions

After the lab assessments are completed, the technology coordinator provides the participant with the appropriate equipment based on their group assignment and functional level. The equipment includes a tablet (ASUS Zenpad 3s 10), a tablet case (Fintie Folio Stand

Cover), stylus, Fitbit Charge 2 HR to monitor physical activity, a set of two pound wrist weights
(SPRI Thumblock Wrist Weight) and an optional stand (Musician's Gear Folding Music Stand).
The technology coordinator then familiarizes participants with the equipment and also guides
participants through the tablet app. Equipment instructions cover basic operation (e.g., turning
the devices on and off, connecting to Wi-Fi, etc.) and maintenance skills (e.g., cleaning the
devices and damage prevention). Regarding app use, participants are carefully guided through
the tablet app content and functions. Following these instructions, participants are then asked to
independently navigate through the app and view all content.

The tablet app includes the foundational elements required for rapid deployment of the
video and textual content, and is password protected to enable detailed usage tracking. The
home page of the app provides brief weekly objectives and links to newly received content.
Other features of the app include a video and article library, calendar for scheduling, and activity
dashboard with Fitbit data. In addition to these features, the M2M^{plus} group's version of the app
includes elements that support social interaction. These ancillary features include the ability to
private-message other users, leaderboards that display user progress, a user profile, newsfeed,
and the ability to comment on and "like" the exercise videos. The app version for the
intervention groups (M2M and M2M^{plus}) includes exercise video content, while the AC group will
only receive access to the infographics and the Fitbit dashboard.

Educational Resources

Each group receives textual content (i.e., articles, infographics) aimed at improving
health through lifestyle changes and tailored to adults with mobility disabilities. This content
includes physical activity recommendations, how to maintain activity, developing healthy
nutrition habits, ways to reduce stress, and other health-related information. Each group
receives this content weekly for 12 weeks (Adoption phase), bi-weekly for the following 12
48 weeks (Transition phase), and monthly for the last 24 weeks (Maintenance phase).

Exercise Video Content

Exercise videos include movements that were adapted from an onsite program that has been conducted at a state-of-the-art fitness facility for people with disabilities. The onsite program incorporated an extensive set of movement routines that were choreographed to the functional needs of people with a range of physical disabilities. For the present study, we modified the onsite program into a video-based package that could be performed independently in the home. The goal of the home-based program is to slowly and safely progress participants towards achieving and maintaining 150 minutes of moderate-intensity physical activity per week, which is based on the guidelines provided by the American College of Sports Medicine.^{29 30}

The exercise intervention is referred to as Movement-2-Music (M2M). The M2M intervention includes three broad sub-programs or levels that can be arranged by program staff to meet the functional needs of the participant: 1) able to use all four limbs; 2) able to use only the upper limbs (e.g., someone with lower extremity paralysis); and 3) able to use only one side of the body to exercise (e.g., individuals with stroke/hemiparesis). The determination of the level for each participant is based on their baseline physical function assessments. Each M2M session includes four sets of exercise routines: flexibility, muscle strength, cardiorespiratory fitness, and balance. The video ends with a cool down breathing routine. Participants are instructed to complete the assigned M2M exercise videos three times per week. The session duration and exercise intensity is gradually increased throughout the program.

The exercise videos are delivered across the three intervention phases: 1) Adoption, weeks 1-12; 2) Transition, weeks 13-24; and 3) Maintenance, weeks 25-48. During the Adoption phase, one new video is uploaded each week to the eHealth platform. The exercise program begins with a total of 15 minutes of exercise at week 1 and slowly progresses to 90 minutes at week 12. The rationale for starting the program at a low duration is to help participants build confidence and gradually incorporate the exercise program into their daily life. Each week the videos include newly added routines that are guided by an M2M instructor and an actor with a

disability that matches the functional level of the participants. When a new routine is introduced,
the first segment of the video shows the M2M instructor breaking down the routine and
the first segment of the video shows the M2M instructor breaking down the routine and
explaining each movement pattern (guided exercise portion). In the following week, the guided
exercise portion of the same routine is removed from the video so that the participants can
perform the routine directly with music. Participants have the ability to view previously
completed exercise videos in their video library.

During the Transition and Maintenance phases, one new video is added biweekly (twice
a month) and monthly, respectively. The goal of the Transition phase is to progress participants
to a goal of 150 minutes of moderate-intensity exercise.^{29 30} The Transition phase starts with
three 30-minute sessions at week 13 of the intervention (week 1 of this phase) and ends with
three 50-minute sessions at week 24 (150 minutes/week). For the Maintenance phase, the
session duration remains the same and there are new monthly videos added as an incentive to
encourage participants to adhere to the study protocol by using new M2M routines.

Social Networking System

A Social Networking System (SNS) is integrated into the eHealth platform for the M2M^{plus}
group. The SNS contains the essential building blocks of social media including (1) identity, (2)
sharing, (3) presence, (4) relationships, (5) reputations, (6) groups, and (7) conversations.³¹
Additionally, M2M^{plus} participants join a weekly video conference group discussion session led
by the project coordinator who has expertise in motivational interviewing strategies. The
purposes of these sessions are to stimulate discussion among participants and to encourage
them to support each other. Each session incorporates behavioral change strategies that have
been modified from previous research with adults with physical disabilities.³² The sessions focus
on topics related to constructs from the Social Cognitive Theory (the theoretical background for
this study): social support, overcoming barriers, and preventing relapse.

Theoretical Framework

The intervention is grounded in the Social Cognitive Theory,³³ which provides a comprehensive, multi-level model of behavioral change emphasizing personal and environmental factors that lead to the adoption and/or maintenance of health promoting behavior.³⁴ According to the Social Cognitive Theory, health behavior change is based on reciprocal determinism of the interplay of 3 domains: (1) the behavior, which involves performance or mastery by the individual (behavioral capability, self-regulation); (2) personal factors that involve cognitive, affective and biological events (e.g., self-efficacy); and (3) the environmental factors that facilitate or impede change (social and physical environmental factors).³⁴ Thus, the intervention includes strategies that aim to increase exercise self-efficacy by including adults with mobility disabilities as demonstrators for exercise videos (M2M and M2M^{plus} only); fostering social support through social networking features (e.g., messages, posts, newsfeeds) and group discussion sessions led by research staff (M2M^{plus} only); and teaching participants self-regulation strategies such as goal setting, journaling, and reinforcements using infographics (all groups).

Attention control group

In addition to the Fitbit, the AC group receives a tablet that only includes the educational materials. After the 48-week assessment is completed, participants receive access to the exercise videos.

Randomization

Eligible participants are randomized into one of three arms with 1:1:1 allocation ratio using a permuted block randomization approach where the block size is unknown to the intervention staff. The randomization sequence is generated a priori using a computer-generated random schedule in a permuted block (SAS version 9.4). The randomization schedule is then embedded into a randomization module in REDCap.²⁸ This system allows

researchers to manage the information with a higher level of security, remove physical envelop,
and set individual level of blinding within the system.

Staff performing recruitment, outcome measurements, and data entry for the primary
outcomes are blinded to group allocation. However, due to the nature of the intervention, it is
not possible to blind the staff administering interventions or the participants. Participants are
instructed not to inform the data collection staff of their intervention status when they return for
follow-up measures.

**Outcome measures**

Lab-based research outcome measures are administered at baseline, 12 weeks, and 48
22 weeks by independent evaluators blind to treatment assignment. Survey measures are
automatically delivered by the REDCap system at baseline, 12 weeks, 24 weeks, and 48 weeks.

The primary outcomes are change in rates of exercise participation and adherence to
the study protocol from baseline to transition (24 weeks). There is an additional follow-up at 48
30 weeks. Exercise data are collected with an activity monitor (Fitbit Charge 2) and stored as total
steps per week,^{35 36} in addition to a self-report measure, the Godin Leisure-Time Exercise
Questionnaire (GLTEQ),³⁷ at the four time points. Adherence is defined as the percentage the
participant meets or exceeds the study protocol's weekly exercise video target minutes, which
begins at 15 minutes for the first week and increases approximately 5 minutes each week until
reaching 150 minutes of exercise video viewing per week by the end of the transition phase.
The 150 minutes per week target will remain throughout the maintenance phase. To improve
adherence, the project coordinator monitors weekly data uploaded from the Fitbit and tablet that
include the date, total steps per day, and the number of minutes spent viewing videos,
respectively. If the Fitbit is inactive for more than three days during the week (7-day period) or if
the amount of video viewing does not meet the prescribed dosage for minutes of activity for that
52 week, the participant is contacted by research staff trained in motivational interviewing

techniques. The purpose of the call is to gather information on non-adherence and aid the
participant in creating an action plan for meeting the prescribed activity.

Health and physical function measures include a battery of tests performed at baseline,
12 weeks, and 48 weeks (see Table 2). In addition, anthropometric measures and vital signs are
assessed at each testing visit. Short forms developed by the National Institutes of Health (NIH)
Patient-Reported Outcome Measurement Information System (PROMIS) are used to measure
symptoms and quality of life indicators. The PROMIS short forms used for this study include
sleep disturbance, pain, depression, anxiety, physical function, and social interactions. Social
cognitive theory measures include surveys and scales on exercise goal-setting, barriers to
physical activity, outcome expectations for exercise, social support, and exercise self-efficacy.
Other measures include app usability, eHealth literacy assessment, and demographics (i.e.,
race, sex, age). A brief tabulation of the measures for this study including information on key
outcome variables, covariates, and time points is shown in Table 2.

[Insert Table 2]

**Power and sample size**

A sample size of 648 participants resulted from conducting conservative sample size
calculations with minimal assumptions which involved the following assumptions: 80% power,
two sided t-test comparing changes in physical activity and adherence, type 1 error rate of 0.025
to account for multiple testing (with two primary outcomes), attrition rate of 0.36, and intent to
treat analyses. Each pairwise comparison (M2M versus AC, M2Mplus versus AC and M2M
versus M2Mplus) is considered as family of hypothesis. We did not account for multiple
comparisons (between different arms) since these are planned a priori. A sample size of 648,
which result in 415 participants as completers after accounting for the 36% attrition, provides
80% power to detect an effect size of 0.375 (Cohen's d) between any of the pairwise
comparisons aforementioned above.

Previous effect sizes (ES) estimated for changes in exercise from two web-based
interventions reported ES of 0.6129 and 0.8, respectively,^{38 39} and ES achieved from two home-
based behavioral interventions to increase exercise in wheelchair users above 0.5.^{40 41} Our
minimum detectable effect size of 0.375 is much smaller than the aforementioned effect sizes to
account for the variability in the effect sizes and winner's curse⁴² (anticipated scenario given that
we aim to recruit a far larger sample size). An assumed attrition rate (AR) of 36% was based on
the study by Froehlich-Grobe et al.⁴¹ While our prior studies had an AR as low as 13%, the
eHealth literature cautions of higher attrition rates. Achieving lower ARs may possibly relate to
the role of health/motivational coaches in the study. Finally, we will be able to bring more
efficiency in terms of power and effect size since our primary statistical analysis is based on
mixed modeling approach. The mixed modeling approach accounts for correlation between
measurements from the same participant that improves that improves power for the same effect
size or provides us the ability to detect a lower effect size for the same power and
aforementioned assumptions.⁴² Mixed modeling technique also utilizes all the data available
(that is imputes any missing outcome data). Hence, despite attrition we will have a larger
sample size while conducting longitudinal analyses, which would again bring more
power/sample efficiency.

38 **Analyses**

40 All statistical analyses are conducted in an intent-to-treat manner, at the individual level.
41 Statistical significance is evaluated at 0.05 levels after adjusting for multiple testing corrections

[revised manuscript text omitted]

	Systems Usability Scale (SUS), Mobile Application Rating Scale (MARS)	A		

¹Measured, ²Self-Report, B=Baseline, end of A=Adoption, T=Transition, M= Maintenance

REFERENCES

1. Lai B CK, Vanerbomb K, Bickel C Scott, Rimmer JH, Motl RW. Characteristics of adults with neurologic disability recruited for exercise trials: a secondary analysis. *Adapted Physical Activity Quarterly* In Press doi: 10.1123/apaq.2017-0109
2. Chen AY, Kim SE, Houtrow AJ, et al. Prevalence of obesity among children with chronic conditions. *Obesity (Silver Spring, Md)* 2010;18(1):210-3. doi: 10.1038/oby.2009.185
3. Fox MH, Witten MH, Lullo C. Reducing Obesity Among People With Disabilities. *Journal of disability policy studies* 2014;25(3):175-85. doi: 10.1177/1044207313494236
4. Froehlich-Grobe K, Lee J, Washburn RA. Disparities in obesity and related conditions among Americans with disabilities. *American journal of preventive medicine* 2013;45(1):83-90. doi: 10.1016/j.amepre.2013.02.021
5. Hsieh K, Rimmer JH, Heller T. Obesity and associated factors in adults with intellectual disability. *Journal of Intellectual Disability Research* 2014;58(9):851-63. doi: 10.1111/jir.12100
6. Liou TH, Pi-Sunyer FX, Laferrere B. Physical disability and obesity. *Nutrition reviews* 2005;63(10):321-31.
7. Rimmer JH, Wang E. Obesity prevalence among a group of Chicago residents with disabilities. *Arch Phys Med Rehabil* 2005;86(7):1461-4.
8. Rimmer JH, Wang E, Yamaki K, et al. Documenting disparities in obesity and disability. SEDL; Austin, TX:2010. FOCUS Technical Brief No. 24.
9. Carroll DD, Courtney-Long EA, Stevens AC, et al. Vital signs: disability and physical activity--United States, 2009-2012. *MMWR Morb Mortal Wkly Rep* 2014;63(18):407-13.
10. Ravesloot C, Seekins T, Young QR. Health Promotion for People with Chronic Illness and Physical Disabilities: The Connection between Health Psychology and Disability Prevention. *Clinical Psychology and Psychotherapy*. 1998;5(2):76-85.
11. Rejeski JW, Focht CB. Aging and Physical Disability: On Integrating Group and Individual Counseling with the Promotion of Physical Activity. *Exercise and Sport Sciences Reviews* 2002;30(4):166-70.
12. Rejeski WJ, Brawley LR, Haskell WL. The prevention challenge: an overview of this supplement. *American journal of preventive medicine* 2003;25(3 Suppl 2):107-9.
13. Rimmer JH, Riley B, Creviston T, et al. Exercise training in a predominantly African-American group of stroke survivors. *Medicine and science in sports and exercise* 2000;32(12):1990-6.
14. Martin Ginis KA, Ma JK, Latimer-Cheung AE, et al. A systematic review of review articles addressing factors related to physical activity participation among children and adults with physical disabilities. *Health psychology review* 2016;10(4):478-94. doi: 10.1080/17437199.2016.1198240
15. Becker H, Stuijbergen A. What Makes It So Hard? Barriers to Health Promotion Experienced by People With Multiple Sclerosis and Polio. *Fam Community Health*. 2004;27(1):75-85.
16. Scelza W, Kalpakjian C, Zemper E, et al. Perceived Barriers to Exercise in People with Spinal Cord Injury. *AM J Phys Rehabil*. 2005;84(8):576-83.
17. Phillips M, Flemming N, Tsintzas K. An exploratory study of physical activity and perceived barriers to exercise in ambulant people with neuromuscular disease compared with unaffected controls. *Clinical Rehabilitation* 2009;23(8):746-55. doi: 10.1177/0269215509334838
18. Zalewski K. Exploring Barriers to Remaining Physically Active: A Case Report of a Person with Multiple Sclerosis. *J Neurol Phys Ther*. 2007;31(1):40-5
19. Clarke P, Ailshire JA, Bader M, et al. Mobility Disability and the Urban Built Environment. *American Journal of Epidemiology* 2008;168(5):506-13. doi: 10.1093/aje/kwn185

20. Vasudevan V, Rimmer JH, Kviz F. Development of the Barriers to Physical Activity Questionnaire for People with Mobility Impairments. *Disability and health journal* 2015;8(4):547-56. doi: 10.1016/j.dhjo.2015.04.007
21. Groah SL, Charlifue S, Tate D, et al. Spinal cord injury and aging: challenges and recommendations for future research. *Am J Phys Med Rehabil* 2012;91(1):80-93. doi: 10.1097/PHM.0b013e31821f70bc
22. Rimmer JH. Exercise and physical activity in persons aging with a physical disability. *Physical medicine and rehabilitation clinics of North America* 2005;16(1):41-56. doi: 10.1016/j.pmr.2004.06.013
23. Webb TL, Joseph J, Yardley L, et al. Using the internet to promote health behavior change: a systematic review and meta-analysis of the impact of theoretical basis, use of behavior change techniques, and mode of delivery on efficacy. *Journal of medical Internet research* 2010;12(1):e4. doi: 10.2196/jmir.1376
24. Cuijpers P, van Straten A, Andersson G. Internet-administered cognitive behavior therapy for health problems: a systematic review. *Journal of Behavioral Medicine* 2008;31(2):169-77. doi: 10.1007/s10865-007-9144-1
25. Chan A, Tetzlaff JM, Altman DG, et al. SPIRIT 2013 statement: Defining standard protocol items for clinical trials. *Annals of Internal Medicine* 2013;158(3):200-07. doi: 10.7326/0003-4819-158-3-201302050-00583
26. Schulz KF, Altman DG, Moher D. CONSORT 2010 statement: updated guidelines for reporting parallel group randomised trials. *BMJ (Clinical research ed)* 2010;340:c332. doi: 10.1136/bmj.c332
27. Lindblad AS, Zingesser P, Sismanyazici-Navaie N. Incentives and barriers to neurological clinical research participation. *Clinical investigation* 2011;1(12):1663-68. doi: 10.4155/cli.11.153
28. Harris PA, Taylor R, Thielke R, et al. Research Electronic Data Capture (REDCap) - A metadata-driven methodology and workflow process for providing translational research informatics support. *Journal of biomedical informatics* 2009;42(2):377-81. doi: 10.1016/j.jbi.2008.08.010
29. Garber CE, Blissmer B, Deschenes MR, et al. American College of Sports Medicine position stand. Quantity and quality of exercise for developing and maintaining cardiorespiratory, musculoskeletal, and neuromotor fitness in apparently healthy adults: guidance for prescribing exercise. *Medicine and science in sports and exercise* 2011;43(7):1334-59. doi: 10.1249/MSS.0b013e318213fefb
30. Ferguson B. ACSM's Guidelines for Exercise Testing and Prescription 9th Ed. 2014. *The Journal of the Canadian Chiropractic Association* 2014;58(3):328-28.
31. Kietzmann JH, Hermkens K, McCarthy IP, et al. Social media? Get serious! Understanding the functional building blocks of social media. *Business Horizons* 2011;54(3):241-51. doi: <https://doi.org/10.1016/j.bushor.2011.01.005>
32. Ellis T, Motl RW. Physical activity behavior change in persons with neurologic disorders: overview and examples from Parkinson disease and multiple sclerosis. *Journal of neurologic physical therapy*. 2013;37(2):85-90. doi: 10.1097/NPT.0b013e31829157c0
33. Bandura A. Social foundations of thought and action: a social cognitive theory. Englewood Cliffs, NJ: Prentice Hall 1986.
34. Bandura A. Health promotion by social cognitive means. *Health education & behavior : the official publication of the Society for Public Health Education* 2004;31(2):143-64. doi: 10.1177/1090198104263660
35. Schrader K, Mentis H, Phipps M, et al. What factors predict Fitbit adherence in Stroke and Parkinson disease? (P6.029). *Neurology* 2017;88(16 Supplement)

36. Gordon R, Bloxham S. Influence of the Fitbit Charge HR on physical activity, aerobic fitness and disability in non-specific back pain participants. *J Sport Med Physical Fitness* 2017;57(12):1669-75. doi: 10.23736/s0022-4707.17.06688-9
37. Godin Leisure-Time Exercise Questionnaire. *Med Sci in Sport Exercise* 1997;29(Supplement):36-38. doi: 10.1097/00005768-199706001-00009
38. Rimmer JH, Rauworth A, Wang E, et al. A randomized controlled trial to increase physical activity and reduce obesity in a predominantly African American group of women with mobility disabilities and severe obesity. *Preventive Medicine* 2009;48(5):473-79. doi: <https://doi.org/10.1016/j.ypmed.2009.02.008>
39. Rimmer J, Wang E, Pellegrini C, et al. Telehealth Weight Management Intervention for Adults with Physical Disabilities A Randomized Controlled Trial. *Am J Phys Med Rehabil*. 2013;92(12):1084-94.
40. Froehlich-Grobe K, White GW. Promoting physical activity among women with mobility impairments: a randomized controlled trial to assess a home- and community-based intervention. *Arch Phys Med Rehabil*. 2004;85(4):640-48. doi: <https://doi.org/10.1016/j.apmr.2003.07.012>
41. Froehlich-Grobe K, Lee J, Aaronson L, et al. Exercise for Everyone: A randomized controlled trial of Project Workout On Wheels in promoting exercise among wheelchair users. *Arch Phys Med Rehabil*. 2014;95(1):20-28. doi: 10.1016/j.apmr.2013.07.006
42. Katherine SB, John PAI, Claire M, et al. Power failure: why small sample size undermines the reliability of neuroscience. *Nature Reviews Neuroscience* 2013;14(5):365. doi: 10.1038/nrn3475
43. P Burnham K, R Anderson D. Multimodel Inference: understanding AIC and BIC in Model Selection. *Sociological Methods and Research*. 2004;33(2)261-304.
44. Burnham K AD. Model selection and multimodel inference: A practical information-theoretic approach. 2 ed: Springer-Verlag 2002.
45. Rimmer JH, Riley B, Wang E, et al. Physical activity participation among persons with disabilities: barriers and facilitators. *American journal of preventive medicine* 2004;26(5):419-25. doi: 10.1016/j.amepre.2004.02.002
46. Salisbury SK, Choy NL, Nitz J. Shoulder pain, range of motion, and functional motor skills after acute tetraplegia. *Arch Phys Med Rehabil* 2003;84(10):1480-5.
47. Rimmer JH. The conspicuous absence of people with disabilities in public fitness and recreation facilities: lack of interest or lack of access? *Am J Health Prom*. 2005;19(5):327-9, ii.
48. Rimmer JH, Padalabalanarayanan S, Malone LA, et al. Fitness facilities still lack accessibility for people with disabilities. *Disability and Health Journal* 2017;10(2):214-21. doi: <https://doi.org/10.1016/j.dhjo.2016.12.011>
49. Cook KF, Molton IR, Jensen MP. Fatigue and aging with a disability. *Arch Phys Med Rehabil* 2011;92(7):1126-33. doi: 10.1016/j.apmr.2011.02.017
50. Miriam W, Martin-Diener E, Bauer G, et al. Comparison of trial participants and open access users of a web-based physical activity intervention regarding adherence, attrition, and repeated participation. *Journal of medical Internet research* 2010;12(1):e3. doi: 10.2196/jmir.1361
51. Jaarsma EA, Dijkstra PU, Geertzen JH, et al. Barriers to and facilitators of sports participation for people with physical disabilities: a systematic review. *Scandinavian journal of medicine & science in sports* 2014;24(6):871-81. doi: 10.1111/sms.12218
52. Rimmer JH, Rubin SS, Braddock D. Barriers to exercise in African American women with physical disabilities. *Arch Phys Med Rehabil* 2000;81(2):182-8.
53. Rimmer JH, Silverman K, Braunschweig C, et al. Feasibility of a health promotion intervention for a group of predominantly African American women with type 2 diabetes. *The Diabetes educator* 2002;28(4):571-80. doi: 10.1177/014572170202800411

54. Rimmer JH, Wang E, Smith D. Barriers associated with exercise and community access for
individuals with stroke. *Journal of rehabilitation research and development*
2008;45(2):315-22.

For peer review only

Figure 1. Study Flowchart

224x321mm (300 x 300 DPI)

SPIRIT 2013 Checklist: Recommended items to address in a clinical trial protocol and related documents*

Section/item	Item No	Description	Addressed on page number
Administrative information			
Title	1	Descriptive title identifying the study design, population, interventions, and, if applicable, trial acronym	1
Trial registration	2a	Trial identifier and registry name. If not yet registered, name of intended registry	1
	2b	All items from the World Health Organization Trial Registration Data Set	1
Protocol version	3	Date and version identifier	informed consent
Funding	4	Sources and types of financial, material, and other support	19
Roles and responsibilities	5a	Names, affiliations, and roles of protocol contributors	title page
	5b	Name and contact information for the trial sponsor	19
	5c	Role of study sponsor and funders, if any, in study design; collection, management, analysis, and interpretation of data; writing of the report; and the decision to submit the report for publication, including whether they will have ultimate authority over any of these activities	18
	5d	Composition, roles, and responsibilities of the coordinating centre, steering committee, endpoint adjudication committee, data management team, and other individuals or groups overseeing the trial, if applicable (see Item 21a for data monitoring committee)	n/a

Introduction

Background and rationale	6a	Description of research question and justification for undertaking the trial, including summary of relevant studies (published and unpublished) examining benefits and harms for each intervention	3-5
	6b	Explanation for choice of comparators	4-5
Objectives	7	Specific objectives or hypotheses	5
Trial design	8	Description of trial design including type of trial (eg, parallel group, crossover, factorial, single group), allocation ratio, and framework (eg, superiority, equivalence, noninferiority, exploratory)	5

Methods: Participants, interventions, and outcomes

Study setting	9	Description of study settings (eg, community clinic, academic hospital) and list of countries where data will be collected. Reference to where list of study sites can be obtained	5-6
Eligibility criteria	10	Inclusion and exclusion criteria for participants. If applicable, eligibility criteria for study centres and individuals who will perform the interventions (eg, surgeons, psychotherapists)	7
Interventions	11a	Interventions for each group with sufficient detail to allow replication, including how and when they will be administered	8-12
	11b	Criteria for discontinuing or modifying allocated interventions for a given trial participant (eg, drug dose change in response to harms, participant request, or improving/worsening disease)	n/a
	11c	Strategies to improve adherence to intervention protocols, and any procedures for monitoring adherence (eg, drug tablet return, laboratory tests)	13
	11d	Relevant concomitant care and interventions that are permitted or prohibited during the trial	7
Outcomes	12	Primary, secondary, and other outcomes, including the specific measurement variable (eg, systolic blood pressure), analysis metric (eg, change from baseline, final value, time to event), method of aggregation (eg, median, proportion), and time point for each outcome. Explanation of the clinical relevance of chosen efficacy and harm outcomes is strongly recommended	13-14; table 2
Participant timeline	13	Time schedule of enrolment, interventions (including any run-ins and washouts), assessments, and visits for participants. A schematic diagram is highly recommended (see Figure)	20

Sample size	14	Estimated number of participants needed to achieve study objectives and how it was determined, including clinical and statistical assumptions supporting any sample size calculations	14-15
Recruitment	15	Strategies for achieving adequate participant enrolment to reach target sample size	5-6
Methods: Assignment of interventions (for controlled trials)
Allocation:
Sequence generation	16a	Method of generating the allocation sequence (eg, computer-generated random numbers), and list of any factors for stratification. To reduce predictability of a random sequence, details of any planned restriction (eg, blocking) should be provided in a separate document that is unavailable to those who enrol participants or assign interventions	12
Allocation concealment mechanism	16b	Mechanism of implementing the allocation sequence (eg, central telephone; sequentially numbered, opaque, sealed envelopes), describing any steps to conceal the sequence until interventions are assigned	11-12
Implementation	16c	Who will generate the allocation sequence, who will enrol participants, and who will assign participants to interventions	8
Blinding (masking)	17a	Who will be blinded after assignment to interventions (eg, trial participants, care providers, outcome assessors, data analysts), and how	12
17b	If blinded, circumstances under which unblinding is permissible, and procedure for revealing a participant's allocated intervention during the trial	n/a
Methods: Data collection, management, and analysis
Data collection methods	18a	Plans for assessment and collection of outcome, baseline, and other trial data, including any related processes to promote data quality (eg, duplicate measurements, training of assessors) and a description of study instruments (eg, questionnaires, laboratory tests) along with their reliability and validity, if known. Reference to where data collection forms can be found, if not in the protocol	13-14; table 2
18b	Plans to promote participant retention and complete follow-up, including list of any outcome data to be collected for participants who discontinue or deviate from intervention protocols	13

Data management	19	Plans for data entry, coding, security, and storage, including any related processes to promote data quality (eg, double data entry; range checks for data values). Reference to where details of data management procedures can be found, if not in the protocol	8
Statistical methods	20a	Statistical methods for analysing primary and secondary outcomes. Reference to where other details of the statistical analysis plan can be found, if not in the protocol	15-16
20b	Methods for any additional analyses (eg, subgroup and adjusted analyses)	15-16
20c	Definition of analysis population relating to protocol non-adherence (eg, as randomised analysis), and any statistical methods to handle missing data (eg, multiple imputation)	15-16
Methods: Monitoring
Data monitoring	21a	Composition of data monitoring committee (DMC); summary of its role and reporting structure; statement of whether it is independent from the sponsor and competing interests; and reference to where further details about its charter can be found, if not in the protocol. Alternatively, an explanation of why a DMC is not needed	16
21b	Description of any interim analyses and stopping guidelines, including who will have access to these interim results and make the final decision to terminate the trial	n/a
Harms	22	Plans for collecting, assessing, reporting, and managing solicited and spontaneously reported adverse events and other unintended effects of trial interventions or trial conduct	15
Auditing	23	Frequency and procedures for auditing trial conduct, if any, and whether the process will be independent from investigators and the sponsor	17
Ethics and dissemination
Research ethics approval	24	Plans for seeking research ethics committee/institutional review board (REC/IRB) approval	15
Protocol amendments	25	Plans for communicating important protocol modifications (eg, changes to eligibility criteria, outcomes, analyses) to relevant parties (eg, investigators, REC/IRBs, trial participants, trial registries, journals, regulators)	15

Consent or assent	26a	Who will obtain informed consent or assent from potential trial participants or authorised surrogates, and how (see Item 32)	14
26b	Additional consent provisions for collection and use of participant data and biological specimens in ancillary studies, if applicable	n/a
Confidentiality	27	How personal information about potential and enrolled participants will be collected, shared, and maintained in order to protect confidentiality before, during, and after the trial	16
Declaration of interests	28	Financial and other competing interests for principal investigators for the overall trial and each study site	19
Access to data	29	Statement of who will have access to the final trial dataset, and disclosure of contractual agreements that limit such access for investigators	16-17
Ancillary and post-trial care	30	Provisions, if any, for ancillary and post-trial care, and for compensation to those who suffer harm from trial participation	informed consent
Dissemination policy	31a	Plans for investigators and sponsor to communicate trial results to participants, healthcare professionals, the public, and other relevant groups (eg, via publication, reporting in results databases, or other data sharing arrangements), including any publication restrictions	16
31b	Authorship eligibility guidelines and any intended use of professional writers	16
31c	Plans, if any, for granting public access to the full protocol, participant-level dataset, and statistical code	n/a
Appendices
Informed consent materials	32	Model consent form and other related documentation given to participants and authorised surrogates	appendix
Biological specimens	33	Plans for collection, laboratory evaluation, and storage of biological specimens for genetic or molecular analysis in the current trial and for future use in ancillary studies, if applicable	n/a

*It is strongly recommended that this checklist be read in conjunction with the SPIRIT 2013 Explanation & Elaboration for important clarification on the items.
 Amendments to the protocol should be tracked and dated. The SPIRIT checklist is copyrighted by the SPIRIT Group under the Creative Commons
 "[Attribution-NonCommercial-NoDerivs 3.0 Unported](https://creativecommons.org/licenses/by-nc-nd/3.0/)" license.

BMJ Open

Rationale and Design of a Scale-Up Project Evaluating Responsiveness to Home Exercise and Lifestyle Tele-Health (SUPER-HEALTH) in People with Mobility Disabilities: A Type 1 Hybrid Design Based Effectiveness Trial

Journal:	BMJ Open
Manuscript ID	bmjopen-2018-023538.R1
Article Type:	Protocol
Date Submitted by the Author:	18-Sep-2018
Complete List of Authors:	Rimmer, James H. ; University of Alabama at Birmingham, UAB/Lakeshore Research Collaborative Mehta, Tapan; University of Alabama at Birmingham, Department of Health Services Administration Wilroy, Jereme; University of Alabama at Birmingham, UAB/Lakeshore Research Collaborative Lai, Byron; University of Alabama at Birmingham, UAB/Lakeshore Research Collaborative Young, Hui-Ju; University of Alabama at Birmingham, UAB/Lakeshore Research Collaborative Kim, Yumi; University of Alabama at Birmingham, UAB/Lakeshore Research Collaborative Pekmezi, D; University of Alabama at Birmingham, Department of Health Behavior Thirumalai, Mohanraj; University of Alabama at Birmingham, Department of Health Services Administration
Primary Subject Heading:	Rehabilitation medicine
Secondary Subject Heading:	Research methods
Keywords:	telerehabilitation, mobility disabilities, telehealth, exercise, disability, mhealth

**Rationale and design of a Scale-Up Project Evaluating Responsiveness to Home Exercise**
**and Lifestyle Tele-Health (SUPER-HEALTH) in people with mobility disabilities: A type 1**
**hybrid design based effectiveness trial**

James Rimmer,¹ PhD, Tapan Mehta,¹ PhD, Jereme Wilroy,¹ PhD, Byron Lai,¹ PhD, Hui-Ju
Young,¹ PhD, Yumi Kim,¹ MA, Dorothy Pekmezi,¹ PhD, Mohanraj Thirumalai,¹ PhD

**Author affiliations**

¹University of Alabama at Birmingham, Birmingham, USA

**Corresponding Author:**

Mohanraj Thirumalai

1720 2nd Avenue South

Birmingham, AL 35294-0113

mohanraj@uab.edu

205-934-7189

**Keywords:** eHealth, Physical Activity, Physically Disabled, mHealth, Humans

**Word count:** 6,643

**ABSTRACT**

**Introduction:** Rates of physical inactivity among people with physical disabilities are
substantially higher than in the general population while access to home-based exercise
programs tailored are almost non-existent. Using a theory-driven eHealth platform, an
innovative exercise program referred to as movement-to-music (M2M) will be delivered as a
customized, home-based exercise intervention for adults with mobility disabilities.

**Methods and analysis:** Participants are being recruited for this type 1 hybrid design based
effectiveness trial through outpatient clinics at a large rehabilitation center and randomized to
one of three groups: a) M2M; b) M2M plus social networking (M2M^{plus}); and c) attention control
(AC). The intervention includes a 12-week adoption phase, 12-week transition phase, and 24-
26 week maintenance phase, at which the collection of objective measures on exercise, fitness and

1 self-reported measures on health will be obtained at the start of each phase and at follow-up.

2 The study compares the effectiveness of M2M and M2M^{plus} in increasing physical activity,
3 adherence, fitness, and physical functioning compared to the AC group and examine the
4 mediators and moderators of the treatment effect.

[revised manuscript text omitted]

To fully address the barriers to accessing exercise, including the need for healthcare
providers to have off-the-shelf resources that they can quickly provide to their patients,
technology can play a substantial role in the delivery of home-based exercise to people with
physical disabilities. Providing exercise opportunities in the comfort of a person's home would
increase the opportunity for regular and sustainable exercise that is currently not available in
most communities across the U.S. The proposed scale up exercise intervention will enroll a
large cohort of adults with mobility disabilities from a university-based outpatient rehabilitation
center to determine if an innovative home-based exercise program can achieve improvements
in physical activity adherence, health and functional outcomes. The study is referred to as

SUPER-HEALTH, which stands for *Scale-Up Project Evaluating Responsiveness to Home*
*Exercise And Lifestyle Tele-Health*.

This study aims to test the effectiveness of a home-based e-Health exercise program for
increasing physical activity among a clinical population of people with mobility disabilities. The
intervention, referred to as Movement-to-Music (M2M), is being compared to an enhanced
version of M2M that includes social networking (M2M^{plus}). Both of these interventions are
compared to an attention control (AC) group. Secondary aims include (1) estimating
improvements in health (pain, sleep, quality of life) and physical function (balance, strength,
endurance) between M2M and M2M^{plus} groups compared to the AC group; and (2) assessing
mediators and moderators of social cognitive theory (self-efficacy, self-regulation, social
support, outcome expectancies) and demographic factors (age, race, disability type) of the
hypothesized treatment effect to understand for whom and how the intervention is effective.

**METHODS**

This protocol was approved by the University's Institutional Review Board (IRB-
160923002) and is registered with ClinicalTrials.gov (#NCT03024320) as a phase III clinical
trial. This paper follows the SPIRIT checklist²⁵, TIDieR checklist²⁶, and conduct and reporting of
the trial follows CONSORT guidelines.²⁷

**Study design**

SUPER-HEALTH is a single site, three-arm parallel group type 1 hybrid design based
effectiveness trial^{28 29} evaluating the effects of two intervention groups (M2M and M2M^{plus})
compared to AC in people with mobility disabilities, with assessments conducted at four time
points: baseline, 12 weeks, 24 weeks, and 48 weeks.

**Recruitment**

Participants are being recruited from outpatient clinics of a large university-based
rehabilitation center through physicians and their staff and by physical mail-outs from January
2018 until the desired sample size is obtained. Supplementary recruitment strategies include

brochures placed in clinics, society events, newsletters, advertisements, and word of mouth.

Additionally, recruitment will include screening more male participants than females.

In order to enhance the likelihood of enrollment through clinician referral,³⁰ research staff
attend periodic meetings with the physicians in the outpatient center. Recruitment from
outpatient clinics include: 1) direct referrals from physicians and their staff; 2) indirect referrals
through advertisement materials in waiting room; 3) targeted in-person visits by research staff
on the day of a patient's clinic appointment; and 4) routine recruitment visits to the waiting room
by research staff. Physicians have been instructed to hand out brochures that include the study
objectives and contact information. The identification of eligible participants for in-person visits
occurs through the review of medical records and clinic appointments using the password-
protected hospital database. After potential participants are identified, the recruitment
coordinator greets the individuals in the waiting room, provides an overview of the study
objectives, and consents individuals interested in participating in the study.

Physical mail-outs are being delivered to potential participants that are identified from
the NIH-funded National Center for Biomedical Computing based at Partners HealthCare
System: Informatics for Integrating Biology and the Bedside (i2b2)
[<https://www.i2b2.org/about/index.html>]. The i2b2 has been used to identify study cohorts within
the outpatient rehabilitation clinics and address research questions by integrating a wide variety
of clinical data sources. After an initial search query of participants located within the same state
where the study is being conducted, and who attend one of the outpatient clinics, we identified
approximately 5,400 people with neuromuscular disorders and musculoskeletal conditions. This
list directs the physical mail-outs, which include a flyer that describes the study and a letter co-
signed by the chair of department overseeing the clinics and the principal investigator. Mail-outs
are being sent in batches of 100 every 2 to 4 weeks and will be increased or decreased
depending on the response rate.

Exclusion criteria

Individuals with a mobility disability are eligible for the study. In order to remain
consistent with other studies,⁴ this was defined as self-reported difficulty (1) walking (some,
much, unable to do) without special equipment use; (2) walking one-quarter of a mile; (3)
remaining on feet for more than two hours; (4) taking 10 uninterrupted steps; (5) kneeling,
stooping, or crouching; or (5) standing up from an armless straight chair. Individuals who meet
the following criteria are not eligible to participate:

- ➤ accumulating more than 60 minutes of moderate/vigorous physical activity per week;
- ➤ do not report having a diagnosis of a physical/mobility disability;
- ➤ not within working age (18 to 64 yrs. of age);
- ➤ currently enrolled in a structured exercise program over the past 6 months;
- ➤ unable to use upper, lower or both sets of extremities to exercise;
- ➤ unable to converse and read English;
- ➤ medically unstable to perform home exercise as determined by their physician;
- ➤ cognitive impairment that may preclude self-directed daily activities
- ➤ no Internet access.

Study Procedures

The study flow diagram includes three phases of the intervention, Adoption (weeks 1 to
12), Transition (weeks 13 to 24), and Maintenance (weeks 25 to 48) (Figure 1). This phased
approach allows for gradual adjustment in the dosage of the intervention and the ability to
capture changes within and across phases. Details of the interventions offered in each arm are
shown in Table 1.

[Insert Figure 1]

[Insert Table 1]

All data storage is established via the Research Electronic Data Capture (REDCap),³¹ an
electronic data capture system. When potential participants fill in their information via the study
website, it is automatically stored in REDCap. Potential participants' information with other
methods of contact (e.g., mail-back, call) is manually entered into REDCap by the recruitment
coordinator. The recruitment coordinator reaches out to potential participants within 48-72 hours
based on their preferred contact methods (i.e., phone, emails) for more information regarding
the study as well as participation eligibility. If eligible, the participant will be consented over the
phone and will then receive baseline surveys to be completed online prior to testing visit. Once
participants complete the baseline testing they will be randomized.

After the group allocation, participants receive a study designated email address which
is uploaded to REDCap. Surveys are automatically sent at the remaining follow-up time points,
12 weeks, 24 weeks, and 48 weeks, to the new email address. There is a 10-day window open
before and after the exact date of follow-up data collection based on the initial lab testing date.
Participants receive the same email a maximum of five times (1 per day) if they do not complete
the survey. When a research staff member notices the missing data after 5 days, the participant
is contacted by phone.

[revised manuscript text omitted]

behavior.³⁷ According to the Social Cognitive Theory, health behavior change is based on
reciprocal determinism of the interplay of 3 domains: (1) the behavior, which involves
performance or mastery by the individual (behavioral capability, self-regulation); (2) personal
factors that involve cognitive, affective and biological events (e.g., self-efficacy); and (3) the
environmental factors that facilitate or impede change (social and physical environmental
factors).³⁷ Thus, the intervention targets exercise self-efficacy through self-regulation strategies
(incremental goal setting, reinforcements using infographics (all groups) and by including adults
with mobility disabilities as demonstrators for exercise videos (M2M and M2M^{plus} only). These
approaches focus on mastery experiences, social modeling and verbal persuasion and thus are
well aligned with Bandura's research on the key influences on self-efficacy.³⁸ Social support for
physical activity intervention studies among individuals without physical disabilities, such as
social networking (e.g., messages, posts, newsfeeds)^{39 40} and group discussion sessions⁴¹ led
by research staff (M2M^{plus} only).

**Attention control group**

In addition to the Fitbit, the AC group receives a tablet that only includes the educational
materials. After the 48-week assessment is completed, participants receive access to the
exercise videos.

**Randomization**

Eligible participants are randomized into one of three arms with 1:1:1 allocation ratio
using a permuted block randomization approach where the block size is unknown to the
intervention staff. The randomization sequence is generated a priori using a computer-
generated random schedule in a permuted block (SAS version 9.4). The randomization
schedule is then embedded into a randomization module in REDCap.³¹ This system allows
researchers to manage the information with a higher level of security, remove physical envelop,
and set individual level of blinding within the system.

Staff performing recruitment, outcome measurements, and data entry for the primary
outcomes are blinded to group allocation. However, due to the nature of the intervention, it is
not possible to blind the staff administering interventions or the participants. Participants are
instructed not to inform the data collection staff of their intervention status when they return for
follow-up measures.

**Intervention Fidelity**

The intervention uses the latest eHealth technology as a means to deliver and monitor
the intervention. Each participant receives the latest in activity monitoring (Fitbit), streaming,
content (M2M and M2M^{plus} only), and a social networking platform (M2M^{plus} only), which paired
with the use of cloud-based technology allows for the intervention to be administered in the
home. Our intervention fidelity plan (Table 2) is based on the *Best Practices and*
*Recommendations from the NIH Behavior Change Consortium (BCC)*. The BCC
recommendations are intended to link theory and application across five primary study phases,

including study design, provider training, monitoring and improving the delivery of the
intervention, and monitoring and improving the enactment of intervention skills.

[Insert Table 2]

**Outcome measures**

Lab-based research outcome measures are administered at baseline, 12 weeks, and 48
6 weeks by independent evaluators blind to treatment assignment. Survey measures are
7 automatically delivered by the REDCap system at baseline, 12 weeks, 24 weeks, and 48 weeks.

The primary outcome is change in rates of exercise participation. Exercise data is
collected with a self-report measure, the Godin Leisure-Time Exercise Questionnaire
(GLTEQ),⁴² at the four time points. A secondary outcome is adherence to the study protocol
from baseline to transition (24 weeks), with an additional follow-up at 48 weeks. Adherence is
defined as the percentage the participant meets or exceeds the study protocol's weekly exercise
session attendance, which is measured by how many times the participant uses the app. The
exercise video begins with 5 minutes of exercise, and each participant is asked to complete
three times each week equaling a total of 15 minutes for the first week. This increases
approximately 5 minutes each week until reaching 150 minutes of exercise video viewing per
17 week (50 minutes of exercise completed three times) by the end of the transition phase. The
18 150 minutes per week will remain throughout the maintenance phase. To improve adherence
and ensure delivery of the intervention, the project coordinator monitors weekly data uploaded
from the Fitbit and tablet that include the date, total steps per day, and a log for the exercise
videos, which includes a timestamp, title of video, and the number of minutes spent viewing. If
there is no Fitbit data for more than seven days or if the amount of logins for video viewing does
not meet the prescribed dosage of three times per week, the participant is contacted by
research staff trained in motivational interviewing techniques. The purpose of the call is to
gather information on non-adherence and aid the participant in creating an action plan for
meeting the prescribed activity.

Other outcomes include health and physical function measures, which involve a battery
of tests performed at baseline, 12 weeks, and 48 weeks (see Table 3). In addition,
anthropometric measures and vital signs are assessed at each testing visit. Short forms
developed by the National Institutes of Health (NIH) Patient-Reported Outcome Measurement
Information System (PROMIS) are used to measure symptoms and quality of life indicators. The
PROMIS short forms used for this study include sleep disturbance, pain, depression, anxiety,
physical function, and social interactions. Social cognitive theory measures include surveys and
scales on exercise goal-setting, barriers to physical activity, outcome expectations for exercise,
social support, and exercise self-efficacy. Other measures include app usability, eHealth literacy
assessment, and demographics (i.e., race, sex, age). A brief tabulation of the measures for this
study including information on key outcome variables, covariates, and time points is shown in
Table 3.

[Insert Table 3]

30 31 14 **Power and sample size**

A sample size of 648 participants resulted from conducting conservative sample size
calculations with minimal assumptions which involved the following assumptions: 80% power,
two sided t-test comparing changes in physical activity, type 1 error rate of 0.025 to account for
multiple testing, attrition rate of 0.36, and intent to treat analyses. Each pairwise comparison
(M2M versus AC, M2Mplus versus AC and M2M versus M2Mplus) is considered as family of
hypothesis. We did not account for multiple comparisons (between different arms) since these
are planned a priori. A sample size of 648, which result in 415 participants as completers after
accounting for the 36% attrition, provides 80% power to detect an effect size of 0.375 (Cohen's
23 d) between any of the pairwise comparisons aforementioned above.

Previous effect sizes (ES) estimated for changes in exercise from two web-based
interventions reported ES of 0.6129 and 0.8, respectively,^{43 44} and ES achieved from two home-
based behavioral interventions to increase exercise in wheelchair users above 0.5.^{45 46} Our

1 minimum detectable effect size of 0.375 is much smaller than the aforementioned effect sizes to

[revised manuscript text omitted]
 & usability	eHealth Literacy Scale (eHEALS) Systems Usability Scale (SUS), Mobile Application Rating Scale (MARS)	B, M A	Covariate

**Author Contributions**

JR provided content for the introduction and discussion. TM provided content for the data
analysis and power calculation sections. MT provided all content on app development and
social networking section. JW compiled the first full draft of the manuscript, providing content for
recruitment section, and made all revisions provided by coauthors. BL provided all content on
equipment and technology. HY provided all content for the exercise video section. YK provided
content on data management protocol. DP provided content for theoretical framework section.
All authors reviewed paper in full and provided revisions on the entire paper.

**Study sponsorship, funding and organization**

This work is supported by a grant from the National Institutes of Health, Eunice Kennedy
Shriver National Institute of Child Health and Human Development (1R01HD085186). This
study is sponsored by the University of Alabama at Birmingham and will be coordinated by the
Lakeshore Foundation, Birmingham, Alabama. The study funder will have no role in the conduct
or evaluation of the trial, nor the authority over the study design, analysis, interpretation of
results, or preparation of manuscripts or abstracts.

**Data Sharing, Patient and Public Involvement, and Competing Interests**

We are committed to the sharing of final research data, being mindful that the rights and
privacy of our research participants must be protected at all times, that there is the need to
protect patentable and other proprietary data (ie, our web-based platform), and that restrictions
on data sharing may be imposed by agreements with third parties. Published data with non-
identifiers will also be shared with the 2018 Physical Activity Guidelines committee upon
request. There was no patient or public involvement in the design of this study. No authors have
any competing interests.

REFERENCES

- 1. Lai B CK, Vanerborn K, Bickel C Scott, Rimmer JH, Motl RW. Characteristics of adults with neurologic
disability recruited for exercise trials: a secondary analysis. *Adapted Physical Activity Quarterly*
In Press doi: 10.1123/apaq.2017-0109
- 2. Chen AY, Kim SE, Houtrow AJ, et al. Prevalence of obesity among children with chronic conditions.
*Obesity (Silver Spring, Md)* 2010;18(1):210-3. doi: 10.1038/oby.2009.185 [published Online First:
2009/06/13]
- 3. Fox MH, Witten MH, Lullo C. Reducing Obesity Among People With Disabilities. *Journal of disability*
*policy studies* 2014;25(3):175-85. doi: 10.1177/1044207313494236
- 4. Froehlich-Grobe K, Lee J, Washburn RA. Disparities in obesity and related conditions among
Americans with disabilities. *Am J Prev Med* 2013;45(1):83-90. doi:
10.1016/j.amepre.2013.02.021 [published Online First: 2013/06/26]
- 5. Hsieh K, Rimmer JH, Heller T. Obesity and associated factors in adults with intellectual disability.
*Journal of Intellectual Disability Research* 2014;58(9):851-63. doi: 10.1111/jir.12100
- 6. Liou TH, Pi-Sunyer FX, Laferrere B. Physical disability and obesity. *Nutrition reviews* 2005;63(10):321-
31. [published Online First: 2005/11/22]
- 7. Rimmer JH, Wang E. Obesity prevalence among a group of Chicago residents with disabilities. *Arch*
*Phys Med Rehabil* 2005;86(7):1461-4. [published Online First: 2005/07/09]
- 8. Rimmer JH, Wang E, Yamaki K, et al. Documenting disparities in obesity and disability2010.
- 9. Carroll DD, Courtney-Long EA, Stevens AC, et al. Vital signs: disability and physical activity--United
States, 2009-2012. *MMWR Morb Mortal Wkly Rep* 2014;63(18):407-13. [published Online First:
2014/05/09]
- 10. Ravesloot C, Seekins T, Young QR. Health Promotion for People with Chronic Illness and Physical
Disabilities: The Connection between Health Psychology and Disability Prevention1998.
- 11. Rejeski JW, Focht CB. Aging and Physical Disability: On Integrating Group and Individual Counseling
with the Promotion of Physical Activity. *Exercise and Sport Sciences Reviews* 2002;30(4):166-70.
- 12. Rejeski WJ, Brawley LR, Haskell WL. The prevention challenge: an overview of this supplement. *Am J*
*Prev Med* 2003;25(3 Suppl 2):107-9. [published Online First: 2003/10/14]
- 13. Rimmer JH, Riley B, Creviston T, et al. Exercise training in a predominantly African-American group of
stroke survivors. *Medicine and science in sports and exercise* 2000;32(12):1990-6. [published
Online First: 2000/12/29]
- 14. Martin Ginis KA, Ma JK, Latimer-Cheung AE, et al. A systematic review of review articles addressing
factors related to physical activity participation among children and adults with physical
disabilities. *Health psychology review* 2016;10(4):478-94. doi: 10.1080/17437199.2016.1198240
[published Online First: 2016/10/30]
- 15. Becker H, Stuifbergen A. What Makes It So Hard? Barriers to Health Promotion Experienced by
People With Multiple Sclerosis and Polio2004.
- 16. Scelza W, Kalpakjian C, Zemper E, et al. Perceived Barriers to Exercise in People with Spinal Cord
Injury2005.
- 17. Phillips M, Flemming N, Tsintzas K. An exploratory study of physical activity and perceived barriers to
exercise in ambulant people with neuromuscular disease compared with unaffected controls.
*Clinical Rehabilitation* 2009;23(8):746-55. doi: 10.1177/0269215509334838
- 18. Zalewski K. Exploring Barriers to Remaining Physically Active: A Case Report of a Person with
Multiple Sclerosis2007.
- 19. Clarke P, Ailshire JA, Bader M, et al. Mobility Disability and the Urban Built Environment. *American*
*Journal of Epidemiology* 2008;168(5):506-13. doi: 10.1093/aje/kwn185

20. Vasudevan V, Rimmer JH, Kviz F. Development of the Barriers to Physical Activity Questionnaire for
People with Mobility Impairments. *Disability and health journal* 2015;8(4):547-56. doi:
10.1016/j.dhjo.2015.04.007
- 21. Groah SL, Charlifue S, Tate D, et al. Spinal cord injury and aging: challenges and recommendations
for future research. *Am J Phys Med Rehabil* 2012;91(1):80-93. doi:
10.1097/PHM.0b013e31821f70bc [published Online First: 2011/06/18]
- 22. Rimmer JH. Exercise and physical activity in persons aging with a physical disability. *Physical
medicine and rehabilitation clinics of North America* 2005;16(1):41-56. doi:
10.1016/j.pmr.2004.06.013 [published Online First: 2004/11/25]
- 23. Webb TL, Joseph J, Yardley L, et al. Using the internet to promote health behavior change: a
systematic review and meta-analysis of the impact of theoretical basis, use of behavior change
techniques, and mode of delivery on efficacy. *Journal of medical Internet research*
2010;12(1):e4. doi: 10.2196/jmir.1376 [published Online First: 2010/02/19]
- 24. Cuijpers P, van Straten A, Andersson G. Internet-administered cognitive behavior therapy for health
problems: a systematic review. *Journal of Behavioral Medicine* 2008;31(2):169-77. doi:
10.1007/s10865-007-9144-1
- 25. Chan A, Tetzlaff JM, Altman DG, et al. Spirit 2013 statement: Defining standard protocol items for
clinical trials. *Annals of Internal Medicine* 2013;158(3):200-07. doi: 10.7326/0003-4819-158-3-
201302050-00583
- 26. Hoffmann TC, Glasziou PP, Boutron I, et al. Better reporting of interventions: template for
intervention description and replication (TIDieR) checklist and guide. *BMJ (Clinical research ed)*
2014;348:g1687. doi: 10.1136/bmj.g1687 [published Online First: 2014/03/13]
- 27. Schulz KF, Altman DG, Moher D. CONSORT 2010 statement: updated guidelines for reporting parallel
group randomised trials. *BMJ (Clinical research ed)* 2010;340:c332. doi: 10.1136/bmj.c332
[published Online First: 2010/03/25]
- 28. Bernet AC, Willens DE, Bauer MS. Effectiveness-implementation hybrid designs: implications for
quality improvement science. *Implementation Science : IS* 2013;8(Suppl 1):S2-S2. doi:
10.1186/1748-5908-8-S1-S2
- 29. Curran GM, Bauer M, Mittman B, et al. Effectiveness-implementation Hybrid Designs: Combining
Elements of Clinical Effectiveness and Implementation Research to Enhance Public Health
Impact. *Medical care* 2012;50(3):217-26. doi: 10.1097/MLR.0b013e3182408812
- 30. Lindblad AS, Zingesser P, Sismanyazici-Navaie N. Incentives and barriers to neurological clinical
research participation. *Clinical investigation* 2011;1(12):1663-68. doi: 10.4155/cli.11.153
- 31. Harris PA, Taylor R, Thielke R, et al. Research Electronic Data Capture (REDCap) - A metadata-driven
methodology and workflow process for providing translational research informatics support.
*Journal of biomedical informatics* 2009;42(2):377-81. doi: 10.1016/j.jbi.2008.08.010
- 32. Garber CE, Blissmer B, Deschenes MR, et al. American College of Sports Medicine position stand.
Quantity and quality of exercise for developing and maintaining cardiorespiratory,
musculoskeletal, and neuromotor fitness in apparently healthy adults: guidance for prescribing
exercise. *Medicine and science in sports and exercise* 2011;43(7):1334-59. doi:
10.1249/MSS.0b013e318213febf [published Online First: 2011/06/23]
- 33. Ferguson B. ACSM's Guidelines for Exercise Testing and Prescription 9th Ed. 2014. *The Journal of the
Canadian Chiropractic Association* 2014;58(3):328-28.
- 34. Kietzmann JH, Hermkens K, McCarthy IP, et al. Social media? Get serious! Understanding the
functional building blocks of social media. *Business Horizons* 2011;54(3):241-51. doi:
<https://doi.org/10.1016/j.bushor.2011.01.005>
- 35. Ellis T, Motl RW. Physical activity behavior change in persons with neurologic disorders: overview
and examples from Parkinson disease and multiple sclerosis. *Journal of neurologic physical
52
60*

1 *therapy* : *JNPT* 2013;37(2):85-90. doi: 10.1097/NPT.0b013e31829157c0 [published Online First:
2 2013/05/02]
3
4 36. Bandura A. Social foundations of thought and action: a social cognitive theory. Englewood Cliffs, NJ:
Prentice Hall 1986.
37. Bandura A. Health promotion by social cognitive means. *Health education & behavior* : the official
*publication of the Society for Public Health Education* 2004;31(2):143-64. doi:
10.1177/1090198104263660 [published Online First: 2004/04/20]
38. Bandura A. Self-efficacy: The exercise of control. New York, NY, US: W H Freeman/Times
Books/Henry Holt & Co. 1997.
39. Maher C, Ferguson M, Vandelanotte C, et al. A Web-Based, Social Networking Physical Activity
Intervention for Insufficiently Active Adults Delivered via Facebook App: Randomized Controlled
Trial. *Journal of medical Internet research* 2015;17(7):e174. doi: 10.2196/jmir.4086 [published
Online First: 2015/07/15]
40. Maher CA, Lewis LK, Ferrar K, et al. Are health behavior change interventions that use online social
networks effective? A systematic review. *Journal of medical Internet research* 2014;16(2):e40.
doi: 10.2196/jmir.2952 [published Online First: 2014/02/20]
41. Befort CA, Donnelly JE, Sullivan DK, et al. Group versus individual phone-based obesity treatment for
rural women. *Eating behaviors* 2010;11(1):11-7. doi: 10.1016/j.eatbeh.2009.08.002 [published
Online First: 2009/12/08]
42. Godin Leisure-Time Exercise Questionnaire. *Medicine& Science in Sports & Exercise* 1997;29(Supplement):36-38. doi: 10.1097/00005768-199706001-00009
43. Rimmer JH, Rauworth A, Wang E, et al. A randomized controlled trial to increase physical activity and
reduce obesity in a predominantly African American group of women with mobility disabilities
and severe obesity. *Preventive Medicine* 2009;48(5):473-79. doi:
<https://doi.org/10.1016/j.yjmed.2009.02.008>
44. Rimmer J, Wang E, Pellegrini C, et al. Telehealth Weight Management Intervention for Adults with
Physical Disabilities A Randomized Controlled Trial 2013.
45. Froehlich-Grobe K, White GW. Promoting physical activity among women with mobility impairments:
a randomized controlled trial to assess a home- and community-based intervention 1 1No
commercial party having a direct financial interest in the results of the research supporting this
article has or will confer a benefit upon the author(s) or upon any organization with which the
author(s) is/are associated. *Archives of Physical Medicine and Rehabilitation* 2004;85(4):640-48.
doi: <https://doi.org/10.1016/j.apmr.2003.07.012>
46. Froehlich-Grobe K, Lee J, Aaronson L, et al. Exercise for Everyone: A randomized controlled trial of
Project Workout On Wheels in promoting exercise among wheelchair users. *Archives of physical*
*medicine and rehabilitation* 2014;95(1):20-28. doi: 10.1016/j.apmr.2013.07.006
47. Katherine SB, John PAI, Claire M, et al. Power failure: why small sample size undermines the
reliability of neuroscience. *Nature Reviews Neuroscience* 2013;14(5):365. doi: 10.1038/nrn3475
48. Bailar JC, Iii, Mosteller F. Guidelines for statistical reporting in articles for medical journals:
Amplifications and explanations. *Annals of Internal Medicine* 1988;108(2):266-73. doi:
10.7326/0003-4819-108-2-266
49. Saville DJ. Multiple Comparison Procedures: The Practical Solution. *The American Statistician*
1990;44(2):174-80. doi: 10.2307/2684163
50. Saville DJ. Multiple Comparison Procedures—Cutting the Gordian Knot. *Agronomy Journal*
2015;107(2):730-35. doi: 10.2134/agronj2012.0394
51. P Burnham K, R Anderson D. Multimodel Inference: understanding AIC and BIC in Model
Selection 2004.

1 52. Burnham K AD. Model selection and multimodel inference: A practical information-theoretic
2 approach. 2 ed: Springer-Verlag 2002.
3
4 53. Rimmer JH, Riley B, Wang E, et al. Physical activity participation among persons with disabilities:
barriers and facilitators. *Am J Prev Med* 2004;26(5):419-25. doi: 10.1016/j.amepre.2004.02.002
[published Online First: 2004/05/29]
54. Salisbury SK, Choy NL, Nitz J. Shoulder pain, range of motion, and functional motor skills after acute
tetraplegia. *Arch Phys Med Rehabil* 2003;84(10):1480-5. [published Online First: 2003/10/31]
55. Rimmer JH. The conspicuous absence of people with disabilities in public fitness and recreation
facilities: lack of interest or lack of access? *American journal of health promotion : AJHP*
2005;19(5):327-9, ii. [published Online First: 2005/05/18]
56. Rimmer JH, Padalabalanarayanan S, Malone LA, et al. Fitness facilities still lack accessibility for
people with disabilities. *Disability and Health Journal* 2017;10(2):214-21. doi:
<https://doi.org/10.1016/j.dhjo.2016.12.011>
57. Cook KF, Molton IR, Jensen MP. Fatigue and aging with a disability. *Arch Phys Med Rehabil*
2011;92(7):1126-33. doi: 10.1016/j.apmr.2011.02.017 [published Online First: 2011/06/28]
58. Miriam W, Martin-Diener E, Bauer G, et al. Comparison of trial participants and open access users of
a web-based physical activity intervention regarding adherence, attrition, and repeated
participation. *Journal of medical Internet research* 2010;12(1):e3. doi: 10.2196/jmir.1361
[published Online First: 2010/02/12]
59. Jaarsma EA, Dijkstra PU, Geertzen JH, et al. Barriers to and facilitators of sports participation for
people with physical disabilities: a systematic review. *Scandinavian journal of medicine & science*
*in sports* 2014;24(6):871-81. doi: 10.1111/sms.12218 [published Online First: 2014/04/16]
60. Rimmer JH, Rubin SS, Braddock D. Barriers to exercise in African American women with physical
disabilities. *Arch Phys Med Rehabil* 2000;81(2):182-8. [published Online First: 2000/02/11]
61. Rimmer JH, Silverman K, Braunschweig C, et al. Feasibility of a health promotion intervention for a
group of predominantly African American women with type 2 diabetes. *The Diabetes educator*
2002;28(4):571-80. doi: 10.1177/014572170202800411 [published Online First: 2002/09/13]
62. Rimmer JH, Wang E, Smith D. Barriers associated with exercise and community access for individuals
with stroke. *Journal of rehabilitation research and development* 2008;45(2):315-22. [published
Online First: 2008/06/21]

Protocol Flowchart

224x321mm (300 x 300 DPI)

The TIDieR (Template for Intervention Description and Replication) Checklist*:

Information to include when describing an intervention and the location of the information

Item number	Item	Where located **	
		Primary paper (page or appendix number)	Other † (details)
	BRIEF NAME		
1.	Provide the name or a phrase that describes the intervention.	Title Page	_____
	WHY		
2.	Describe any rationale, theory, or goal of the elements essential to the intervention.	1-5	_____
	WHAT		
3.	Materials: Describe any physical or informational materials used in the intervention, including those provided to participants or used in intervention delivery or in training of intervention providers.	8-9	_____
	Provide information on where the materials can be accessed (e.g. online appendix, URL).		
4.	Procedures: Describe each of the procedures, activities, and/or processes used in the intervention, including any enabling or support activities.	10-11	_____
	WHO PROVIDED		
5.	For each category of intervention provider (e.g. psychologist, nursing assistant), describe their expertise, background and any specific training given.	11-12	_____
	HOW		
6.	Describe the modes of delivery (e.g. face-to-face or by some other mechanism, such as internet or telephone) of the intervention and whether it was provided individually or in a group.	10-12	_____
	WHERE		
7.	Describe the type(s) of location(s) where the intervention occurred, including any necessary infrastructure or relevant features.	Title, 18	_____

TIDieR checklist

WHEN and HOW MUCH

8. Describe the number of times the intervention was delivered and over what period of time including the number of sessions, their schedule, and their duration, intensity or dose.

9-11

TAILORING

9. If the intervention was planned to be personalised, titrated or adapted, then describe what, why, when, and how.

10

MODIFICATIONS

10.† If the intervention was modified during the course of the study, describe the changes (what, why, when, and how).

N/A

HOW WELL

11. Planned: If intervention adherence or fidelity was assessed, describe how and by whom, and if any strategies were used to maintain or improve fidelity, describe them.

13

12.‡ Actual: If intervention adherence or fidelity was assessed, describe the extent to which the intervention was delivered as planned.

N/A

** **Authors** - use N/A if an item is not applicable for the intervention being described. **Reviewers** – use ‘?’ if information about the element is not reported/not sufficiently reported.

† If the information is not provided in the primary paper, give details of where this information is available. This may include locations such as a published protocol or other published papers (provide citation details) or a website (provide the URL).

‡ If completing the TIDieR checklist for a protocol, these items are not relevant to the protocol and cannot be described until the study is complete.

* We strongly recommend using this checklist in conjunction with the TIDieR guide (see *BMJ* 2014;348:g1687) which contains an explanation and elaboration for each item.

* The focus of TIDieR is on reporting details of the intervention elements (and where relevant, comparison elements) of a study. Other elements and methodological features of studies are covered by other reporting statements and checklists and have not been duplicated as part of the TIDieR checklist. When a **randomised trial** is being reported, the TIDieR checklist should be used in conjunction with the CONSORT statement (see www.consort-statement.org) as an extension of **Item 5 of the CONSORT 2010 Statement**. When a **clinical trial protocol** is being reported, the TIDieR checklist should be used in conjunction with the SPIRIT statement as an extension of **Item 11 of the SPIRIT 2013 Statement** (see www.spirit-statement.org). For alternate study designs, TIDieR can be used in conjunction with the appropriate checklist for that study design (see www.equator-network.org).

TIDieR checklist

SPIRIT 2013 Checklist: Recommended items to address in a clinical trial protocol and related documents*

Section/item	Item No	Description	Addressed on page number
Administrative information			
Title	1	Descriptive title identifying the study design, population, interventions, and, if applicable, trial acronym	1
Trial registration	2a	Trial identifier and registry name. If not yet registered, name of intended registry	1
	2b	All items from the World Health Organization Trial Registration Data Set	1
Protocol version	3	Date and version identifier	informed consent
Funding	4	Sources and types of financial, material, and other support	19
Roles and responsibilities	5a	Names, affiliations, and roles of protocol contributors	title page
	5b	Name and contact information for the trial sponsor	19
	5c	Role of study sponsor and funders, if any, in study design; collection, management, analysis, and interpretation of data; writing of the report; and the decision to submit the report for publication, including whether they will have ultimate authority over any of these activities	18
	5d	Composition, roles, and responsibilities of the coordinating centre, steering committee, endpoint adjudication committee, data management team, and other individuals or groups overseeing the trial, if applicable (see Item 21a for data monitoring committee)	n/a

Introduction

Background and rationale	6a	Description of research question and justification for undertaking the trial, including summary of relevant studies (published and unpublished) examining benefits and harms for each intervention	3-5
	6b	Explanation for choice of comparators	4-5
Objectives	7	Specific objectives or hypotheses	5
Trial design	8	Description of trial design including type of trial (eg, parallel group, crossover, factorial, single group), allocation ratio, and framework (eg, superiority, equivalence, noninferiority, exploratory)	5

Methods: Participants, interventions, and outcomes

Study setting	9	Description of study settings (eg, community clinic, academic hospital) and list of countries where data will be collected. Reference to where list of study sites can be obtained	5-6
Eligibility criteria	10	Inclusion and exclusion criteria for participants. If applicable, eligibility criteria for study centres and individuals who will perform the interventions (eg, surgeons, psychotherapists)	7
Interventions	11a	Interventions for each group with sufficient detail to allow replication, including how and when they will be administered	8-12
	11b	Criteria for discontinuing or modifying allocated interventions for a given trial participant (eg, drug dose change in response to harms, participant request, or improving/worsening disease)	n/a
	11c	Strategies to improve adherence to intervention protocols, and any procedures for monitoring adherence (eg, drug tablet return, laboratory tests)	13
	11d	Relevant concomitant care and interventions that are permitted or prohibited during the trial	7
Outcomes	12	Primary, secondary, and other outcomes, including the specific measurement variable (eg, systolic blood pressure), analysis metric (eg, change from baseline, final value, time to event), method of aggregation (eg, median, proportion), and time point for each outcome. Explanation of the clinical relevance of chosen efficacy and harm outcomes is strongly recommended	13-14; table 3
Participant timeline	13	Time schedule of enrolment, interventions (including any run-ins and washouts), assessments, and visits for participants. A schematic diagram is highly recommended (see Figure)	20

Sample size	14	Estimated number of participants needed to achieve study objectives and how it was determined, including clinical and statistical assumptions supporting any sample size calculations	14-15
Recruitment	15	Strategies for achieving adequate participant enrolment to reach target sample size	5-6
Methods: Assignment of interventions (for controlled trials)
Allocation:
Sequence generation	16a	Method of generating the allocation sequence (eg, computer-generated random numbers), and list of any factors for stratification. To reduce predictability of a random sequence, details of any planned restriction (eg, blocking) should be provided in a separate document that is unavailable to those who enrol participants or assign interventions	12
Allocation concealment mechanism	16b	Mechanism of implementing the allocation sequence (eg, central telephone; sequentially numbered, opaque, sealed envelopes), describing any steps to conceal the sequence until interventions are assigned	11-12
Implementation	16c	Who will generate the allocation sequence, who will enrol participants, and who will assign participants to interventions	8
Blinding (masking)	17a	Who will be blinded after assignment to interventions (eg, trial participants, care providers, outcome assessors, data analysts), and how	12
17b	If blinded, circumstances under which unblinding is permissible, and procedure for revealing a participant's allocated intervention during the trial	n/a
Methods: Data collection, management, and analysis
Data collection methods	18a	Plans for assessment and collection of outcome, baseline, and other trial data, including any related processes to promote data quality (eg, duplicate measurements, training of assessors) and a description of study instruments (eg, questionnaires, laboratory tests) along with their reliability and validity, if known. Reference to where data collection forms can be found, if not in the protocol	13-14; table 3
18b	Plans to promote participant retention and complete follow-up, including list of any outcome data to be collected for participants who discontinue or deviate from intervention protocols	13

Data management	19	Plans for data entry, coding, security, and storage, including any related processes to promote data quality (eg, double data entry; range checks for data values). Reference to where details of data management procedures can be found, if not in the protocol	8
Statistical methods	20a	Statistical methods for analysing primary and secondary outcomes. Reference to where other details of the statistical analysis plan can be found, if not in the protocol	15-16
	20b	Methods for any additional analyses (eg, subgroup and adjusted analyses)	15-16
	20c	Definition of analysis population relating to protocol non-adherence (eg, as randomised analysis), and any statistical methods to handle missing data (eg, multiple imputation)	15-16
Methods: Monitoring			
Data monitoring	21a	Composition of data monitoring committee (DMC); summary of its role and reporting structure; statement of whether it is independent from the sponsor and competing interests; and reference to where further details about its charter can be found, if not in the protocol. Alternatively, an explanation of why a DMC is not needed	16
	21b	Description of any interim analyses and stopping guidelines, including who will have access to these interim results and make the final decision to terminate the trial	n/a
Harms	22	Plans for collecting, assessing, reporting, and managing solicited and spontaneously reported adverse events and other unintended effects of trial interventions or trial conduct	15
Auditing	23	Frequency and procedures for auditing trial conduct, if any, and whether the process will be independent from investigators and the sponsor	17
Ethics and dissemination			
Research ethics approval	24	Plans for seeking research ethics committee/institutional review board (REC/IRB) approval	15
Protocol amendments	25	Plans for communicating important protocol modifications (eg, changes to eligibility criteria, outcomes, analyses) to relevant parties (eg, investigators, REC/IRBs, trial participants, trial registries, journals, regulators)	15

Consent or assent	26a	Who will obtain informed consent or assent from potential trial participants or authorised surrogates, and how (see Item 32)	14
26b	Additional consent provisions for collection and use of participant data and biological specimens in ancillary studies, if applicable	n/a
Confidentiality	27	How personal information about potential and enrolled participants will be collected, shared, and maintained in order to protect confidentiality before, during, and after the trial	16
Declaration of interests	28	Financial and other competing interests for principal investigators for the overall trial and each study site	19
Access to data	29	Statement of who will have access to the final trial dataset, and disclosure of contractual agreements that limit such access for investigators	16-17
Ancillary and post-trial care	30	Provisions, if any, for ancillary and post-trial care, and for compensation to those who suffer harm from trial participation	informed consent
Dissemination policy	31a	Plans for investigators and sponsor to communicate trial results to participants, healthcare professionals, the public, and other relevant groups (eg, via publication, reporting in results databases, or other data sharing arrangements), including any publication restrictions	16
31b	Authorship eligibility guidelines and any intended use of professional writers	16
31c	Plans, if any, for granting public access to the full protocol, participant-level dataset, and statistical code	n/a
Appendices
Informed consent materials	32	Model consent form and other related documentation given to participants and authorised surrogates	appendix
Biological specimens	33	Plans for collection, laboratory evaluation, and storage of biological specimens for genetic or molecular analysis in the current trial and for future use in ancillary studies, if applicable	n/a

*It is strongly recommended that this checklist be read in conjunction with the SPIRIT 2013 Explanation & Elaboration for important clarification on the items.
 Amendments to the protocol should be tracked and dated. The SPIRIT checklist is copyrighted by the SPIRIT Group under the Creative Commons
 "[Attribution-NonCommercial-NoDerivs 3.0 Unported](https://creativecommons.org/licenses/by-nc-nd/3.0/)" license.

BMJ Open

Rationale and Design of a Scale-Up Project Evaluating Responsiveness to Home Exercise and Lifestyle Tele-Health (SUPER-HEALTH) in People with Mobility Disabilities: A Type 1 Hybrid Design Based Effectiveness Trial

Journal:	BMJ Open
Manuscript ID	bmjopen-2018-023538.R2
Article Type:	Protocol
Date Submitted by the Author:	29-Nov-2018
Complete List of Authors:	Rimmer, James H. ; University of Alabama at Birmingham, UAB/Lakeshore Research Collaborative Mehta, Tapan; University of Alabama at Birmingham, Department of Health Services Administration Wilroy, Jereme; University of Alabama at Birmingham, UAB/Lakeshore Research Collaborative Lai, Byron; University of Alabama at Birmingham, UAB/Lakeshore Research Collaborative Young, Hui-Ju; University of Alabama at Birmingham, UAB/Lakeshore Research Collaborative Kim, Yumi; University of Alabama at Birmingham, UAB/Lakeshore Research Collaborative Pekmezi, D; University of Alabama at Birmingham, Department of Health Behavior Thirumalai, Mohanraj; University of Alabama at Birmingham, Department of Health Services Administration
Primary Subject Heading:	Rehabilitation medicine
Secondary Subject Heading:	Research methods
Keywords:	telerehabilitation, mobility disabilities, telehealth, exercise, disability, mhealth

**Rationale and design of a Scale-Up Project Evaluating Responsiveness to Home Exercise**
**and Lifestyle Tele-Health (SUPER-HEALTH) in people with mobility disabilities: A type 1**
**hybrid design based effectiveness trial**

James Rimmer,¹ PhD, Tapan Mehta,¹ PhD, Jereme Wilroy,¹ PhD, Byron Lai,¹ PhD, Hui-Ju
Young,¹ PhD, Yumi Kim,¹ MA, Dorothy Pekmezi,¹ PhD, Mohanraj Thirumalai,¹ PhD

**Author affiliations**

¹University of Alabama at Birmingham, Birmingham, Alabama

**Author Contributions**

JR provided content for the introduction and discussion. TM provided content for the data
analysis and power calculation sections. MT provided all content on app development and
social networking section. JW compiled the first full draft of the manuscript, providing content for
recruitment section, and made all revisions provided by coauthors. BL provided all content on
equipment and technology. HY provided all content for the exercise video section. YK provided
content on data management protocol. DP provided content for theoretical framework section.
All authors reviewed paper in full and provided revisions on the entire paper.

**Keywords:** eHealth, Physical Activity, Physically Disabled, mHealth, Humans

**Word count:** 6,137

**Corresponding Author:**

Mohanraj Thirumalai

1720 2nd Avenue South

Birmingham, AL 35294-0113

mohanraj@uab.edu

205-934-7189

**ABSTRACT**

**Introduction:** Rates of physical inactivity among people with physical disabilities are
substantially higher than in the general population while access to home-based exercise
programs tailored are almost non-existent. Using a theory-driven eHealth platform, an

1 innovative exercise program referred to as movement-to-music (M2M) will be delivered as a
2 customized, home-based exercise intervention for adults with mobility disabilities.

[revised manuscript text omitted]

This primary aim of this study is to test the effectiveness of a home-based e-Health
exercise program for increasing physical activity among a clinical population of people with
mobility disabilities. The intervention, referred to as Movement-to-Music (M2M), is being
compared to an enhanced version of M2M that includes social networking (M2M^{plus}). Both of
these interventions are compared to an attention control (AC) group. Secondary aims include
those related to intervention effectiveness and implementation. Regarding effectiveness, aims
include (1) estimating improvements in health (pain, sleep, quality of life) and physical function
(balance, strength, endurance) between M2M and M2M^{plus} groups compared to the AC group;
and (2) assessing mediators and moderators of social cognitive theory (self-efficacy, self-
regulation, social support, outcome expectancies) and demographic factors (age, race, disability
type) of the hypothesized treatment effect to understand for whom and how the intervention is
effective. Implementation is assessed via participant uptake of the intervention, whereby the aim
is to explore participant flow throughout all stages of the study (contact through enrollment and
intervention adoption through intervention maintenance).

**METHODS**

This protocol was approved by the University's Institutional Review Board (IRB-
160923002) and is registered with ClinicalTrials.gov (#NCT03024320) as a phase III clinical
trial. This paper follows the SPIRIT checklist²⁷, TIDieR checklist²⁸, and conduct and reporting of
the trial follows CONSORT guidelines.²⁹

**Study design**

SUPER-HEALTH is a single site, three-arm parallel group type 1 hybrid design based
effectiveness trial^{30 31} evaluating the effects of two intervention groups (M2M and M2M^{plus})
compared to AC in people with mobility disabilities, with assessments conducted at four time
points: baseline, 12 weeks, 24 weeks, and 48 weeks.

**Recruitment**

Participants are being recruited from outpatient clinics of a large university-based
rehabilitation center through physicians and their staff and by physical mail-outs from January
2018 until the desired sample size is obtained. Supplementary recruitment strategies include
brochures placed in clinics, society events, newsletters, advertisements, and word of mouth.
Additionally, recruitment will include screening more male participants than females.

In order to enhance the likelihood of enrollment through clinician referral,³² research staff
attend periodic meetings with the physicians in the outpatient center. Recruitment from
outpatient clinics include: 1) direct referrals from physicians and their staff; 2) indirect referrals
through advertisement materials in waiting room; 3) targeted in-person visits by research staff
on the day of a patient's clinic appointment; and 4) routine recruitment visits to the waiting room
by research staff. Physicians have been instructed to hand out brochures that include the study
objectives and contact information. The identification of eligible participants for in-person visits
occurs through the review of medical records and clinic appointments using the password-
protected hospital database. After potential participants are identified, the recruitment
coordinator greets the individuals in the waiting room, provides an overview of the study
objectives, and consents individuals interested in participating in the study.

Physical mail-outs are being delivered to potential participants that are identified from
the NIH-funded National Center for Biomedical Computing based at Partners HealthCare
System: Informatics for Integrating Biology and the Bedside (i2b2)
[<https://www.i2b2.org/about/index.html>]. The i2b2 has been used to identify study cohorts within
the outpatient rehabilitation clinics and address research questions by integrating a wide variety
of clinical data sources. After an initial search query of participants located within the same state
where the study is being conducted, and who attend one of the outpatient clinics, we identified
approximately 5,400 people with neuromuscular disorders and musculoskeletal conditions. This
list directs the physical mail-outs, which include a flyer that describes the study and a letter co-
signed by the chair of department overseeing the clinics and the principal investigator. Mail-outs

are being sent in batches of 100 every 2 to 4 weeks and will be increased or decreased
depending on the response rate.

**Exclusion criteria**

Individuals with a mobility disability are eligible for the study. In order to remain
consistent with other studies,⁴ this was defined as self-reported difficulty (1) walking (some,
much, unable to do) without special equipment use; (2) walking one-quarter of a mile; (3)
remaining on feet for more than two hours; (4) taking 10 uninterrupted steps; (5) kneeling,
stooping, or crouching; or (5) standing up from an armless straight chair. Individuals who meet
the following criteria are not eligible to participate:

- ➤ accumulating more than 60 minutes of moderate/vigorous physical activity per week;
- ➤ do not report having a diagnosis of a physical/mobility disability;
- ➤ not within working age (18 to 64 yrs. of age);
- ➤ currently enrolled in a structured exercise program over the past 6 months;
- ➤ unable to use upper, lower or both sets of extremities to exercise;
- ➤ unable to converse and read English;
- ➤ medically unstable to perform home exercise as determined by their physician;
- ➤ cognitive impairment that may preclude self-directed daily activities
- ➤ no Wi-Fi Internet access.

**Study Procedures**

The study flow diagram includes three phases of the intervention, Adoption (weeks 1 to
12), Transition (weeks 13 to 24), and Maintenance (weeks 25 to 48) (Figure 1). This phased
approach allows for gradual adjustment in the dosage of the intervention and the ability to
capture changes within and across phases. Details of the interventions offered in each arm are
shown in Table 1.

[Insert Figure 1]

[Insert Table 1]

All data storage is established via the Research Electronic Data Capture (REDCap),³³ an
electronic data capture system. When potential participants fill in their information via the study
website, it is automatically stored in REDCap. Potential participants' information with other
methods of contact (e.g., mail-back, call) is manually entered into REDCap by the recruitment
coordinator. The recruitment coordinator reaches out to potential participants within 48-72 hours
based on their preferred contact methods (i.e., phone, emails) for more information regarding
the study as well as participation eligibility. If eligible, the participant will be consented over the
phone and will then receive baseline surveys to be completed online prior to testing visit. Once
participants complete the baseline testing they will be randomized.

After the group allocation, participants receive a study designated email address which
is uploaded to REDCap. Surveys are automatically sent at the remaining follow-up time points,
12 weeks, 24 weeks, and 48 weeks, to the new email address. There is a 10-day window open
before and after the exact date of follow-up data collection based on the initial lab testing date.
Participants receive the same email a maximum of five times (1 per day) if they do not complete
the survey. When a research staff member notices the missing data after 5 days, the participant
is contacted by phone.

[revised manuscript text omitted]

behavior.³⁹ According to the Social Cognitive Theory, health behavior change is based on
reciprocal determinism of the interplay of 3 domains: (1) the behavior, which involves
performance or mastery by the individual (behavioral capability, self-regulation); (2) personal
factors that involve cognitive, affective and biological events (e.g., self-efficacy); and (3) the
environmental factors that facilitate or impede change (social and physical environmental
factors).³⁹ Thus, the intervention targets exercise self-efficacy through self-regulation strategies
(incremental goal setting, reinforcements using infographics (all groups) and by including adults
with mobility disabilities as demonstrators for exercise videos (M2M and M2M^{plus} only). These
approaches focus on mastery experiences, social modeling and verbal persuasion and thus are
well aligned with Bandura's research on the key influences on self-efficacy.⁴⁰ Social support for
physical activity intervention studies among individuals without physical disabilities, such as
social networking (e.g., messages, posts, newsfeeds)^{41 42} and group discussion sessions⁴³ led
by research staff (M2M^{plus} only).

**Attention control group**

In addition to the Fitbit, the AC group receives a tablet that only includes the educational
materials. After the 48-week assessment is completed, participants receive access to the
exercise videos.

**Randomization**

Eligible participants are randomized into one of three arms with 1:1:1 allocation ratio
using a permuted block randomization approach where the block size is unknown to the
intervention staff. The randomization sequence is generated a priori using a computer-
generated random schedule in a permuted block (SAS version 9.4). The randomization
schedule is then embedded into a randomization module in REDCap.³³ This system allows
researchers to manage the information with a higher level of security, remove physical envelop,
and set individual level of blinding within the system.

[revised manuscript text omitted]

**Power and sample size**

A sample size of 648 participants resulted from conducting conservative sample size
calculations with minimal assumptions, which involved the following assumptions: 80% power,
two sided t-test comparing changes in the primary outcome of physical activity, family-wise error
rate of 0.05 to account for multiple testing, attrition rate of 36%, and intent to treat analyses.
Each pairwise comparison for the primary outcome (M2M versus AC, M2Mplus versus AC and
M2M versus M2Mplus) is considered as a family of hypotheses. We did not account for multiple
comparisons (between different arms) since these are planned a priori. A sample size of 648,

which result in 415 participants as completers after accounting for the 36% attrition, provides
90% power to detect an effect size of 0.32 (Cohen's d) between any of the pairwise
comparisons aforementioned above.

We can consider the 2 sets of secondary outcomes (4 objective measures and 4 patient
reported measures – see Table 3) as separate families of hypotheses. Again, we do not account
for multiple comparisons (pairwise comparison between 3 arms) but do account for multiple
outcomes testing within each pairwise comparison. Hence, assuming a type 1 error rate of
0.0125 with a conservative Bonferroni correction for multiple testing, we will have 90% power to
detect an effect size of 0.373.

Previous effect sizes (ES) estimated for changes in exercise from two web-based
interventions reported ES of 0.6129 and 0.8, respectively,^{45 46} and ES achieved from two home-
based behavioral interventions to increase exercise in wheelchair users above 0.5.^{47 48} Our
minimum detectable effect size of 0.375 is much smaller than the aforementioned effect sizes to

[revised manuscript text omitted]
 & usability	eHealth Literacy Scale (eHEALS) Systems Usability Scale (SUS), Mobile Application Rating Scale (MARS)	B, M A	Covariate

REFERENCES

- 1. Lai B CK, Vanerborn K, Bickel C Scott, Rimmer JH, Motl RW. Characteristics of adults with neurologic
disability recruited for exercise trials: a secondary analysis. *Adapted Physical Activity Quarterly*
In Press doi: 10.1123/apaq.2017-0109
- 2. Chen AY, Kim SE, Houtrow AJ, et al. Prevalence of obesity among children with chronic conditions.
*Obesity (Silver Spring, Md)* 2010;18(1):210-3. doi: 10.1038/oby.2009.185 [published Online First:
2009/06/13]
- 3. Fox MH, Witten MH, Lullo C. Reducing Obesity Among People With Disabilities. *Journal of disability*
*policy studies* 2014;25(3):175-85. doi: 10.1177/1044207313494236
- 4. Froehlich-Grobe K, Lee J, Washburn RA. Disparities in obesity and related conditions among
Americans with disabilities. *Am J Prev Med* 2013;45(1):83-90. doi:
10.1016/j.amepre.2013.02.021 [published Online First: 2013/06/26]
- 5. Hsieh K, Rimmer JH, Heller T. Obesity and associated factors in adults with intellectual disability.
*Journal of Intellectual Disability Research* 2014;58(9):851-63. doi: 10.1111/jir.12100
- 6. Liou TH, Pi-Sunyer FX, Laferrere B. Physical disability and obesity. *Nutrition reviews* 2005;63(10):321-
31. [published Online First: 2005/11/22]
- 7. Rimmer JH, Wang E. Obesity prevalence among a group of Chicago residents with disabilities. *Arch*
*Phys Med Rehabil* 2005;86(7):1461-4. [published Online First: 2005/07/09]
- 8. Rimmer JH, Wang E, Yamaki K, et al. Documenting disparities in obesity and disability2010.
- 9. Carroll DD, Courtney-Long EA, Stevens AC, et al. Vital signs: disability and physical activity--United
States, 2009-2012. *MMWR Morb Mortal Wkly Rep* 2014;63(18):407-13. [published Online First:
2014/05/09]
- 10. Ravesloot C, Seekins T, Young QR. Health Promotion for People with Chronic Illness and Physical
Disabilities: The Connection between Health Psychology and Disability Prevention1998.
- 11. Rejeski JW, Focht CB. Aging and Physical Disability: On Integrating Group and Individual Counseling
with the Promotion of Physical Activity. *Exercise and Sport Sciences Reviews* 2002;30(4):166-70.
- 12. Rejeski WJ, Brawley LR, Haskell WL. The prevention challenge: an overview of this supplement. *Am J*
*Prev Med* 2003;25(3 Suppl 2):107-9. [published Online First: 2003/10/14]
- 13. Rimmer JH, Riley B, Creviston T, et al. Exercise training in a predominantly African-American group of
stroke survivors. *Medicine and science in sports and exercise* 2000;32(12):1990-6. [published
Online First: 2000/12/29]
- 14. Martin Ginis KA, Ma JK, Latimer-Cheung AE, et al. A systematic review of review articles addressing
factors related to physical activity participation among children and adults with physical
disabilities. *Health psychology review* 2016;10(4):478-94. doi: 10.1080/17437199.2016.1198240
[published Online First: 2016/10/30]
- 15. Becker H, Stuifbergen A. What Makes It So Hard? Barriers to Health Promotion Experienced by
People With Multiple Sclerosis and Polio2004.
- 16. Scelza W, Kalpakjian C, Zemper E, et al. Perceived Barriers to Exercise in People with Spinal Cord
Injury2005.
- 17. Phillips M, Flemming N, Tsintzas K. An exploratory study of physical activity and perceived barriers to
exercise in ambulant people with neuromuscular disease compared with unaffected controls.
*Clinical Rehabilitation* 2009;23(8):746-55. doi: 10.1177/0269215509334838
- 18. Zalewski K. Exploring Barriers to Remaining Physically Active: A Case Report of a Person with
Multiple Sclerosis2007.
- 19. Clarke P, Ailshire JA, Bader M, et al. Mobility Disability and the Urban Built Environment. *American*
*Journal of Epidemiology* 2008;168(5):506-13. doi: 10.1093/aje/kwn185

1 20. Vasudevan V, Rimmer JH, Kviz F. Development of the Barriers to Physical Activity Questionnaire for
2 People with Mobility Impairments. *Disability and health journal* 2015;8(4):547-56. doi:
3 10.1016/j.dhjo.2015.04.007
4
4 21. Groah SL, Charlifue S, Tate D, et al. Spinal cord injury and aging: challenges and recommendations
5 for future research. *Am J Phys Med Rehabil* 2012;91(1):80-93. doi:
6 10.1097/PHM.0b013e31821f70bc [published Online First: 2011/06/18]
7
22. Rimmer JH. Exercise and physical activity in persons aging with a physical disability. *Physical
medicine and rehabilitation clinics of North America* 2005;16(1):41-56. doi:
10.1016/j.pmr.2004.06.013 [published Online First: 2004/11/25]
23. Webb TL, Joseph J, Yardley L, et al. Using the internet to promote health behavior change: a
systematic review and meta-analysis of the impact of theoretical basis, use of behavior change
techniques, and mode of delivery on efficacy. *Journal of medical Internet research*
2010;12(1):e4. doi: 10.2196/jmir.1376 [published Online First: 2010/02/19]
24. Cuijpers P, van Straten A, Andersson G. Internet-administered cognitive behavior therapy for health
problems: a systematic review. *Journal of Behavioral Medicine* 2008;31(2):169-77. doi:
10.1007/s10865-007-9144-1
25. Rimmer JH, Lai B, Young HJ. Bending the Arc of Exercise and Recreation Technology Toward People
With Disabilities. *Arch Phys Med Rehabil* 2016;97(9 Suppl):S247-51. doi:
10.1016/j.apmr.2016.02.029 [published Online First: 2016/06/09]
26. Lai B, Kim Y, Wilroy J, et al. Sustainability of exercise intervention outcomes among people with
disabilities: a secondary review. *Disabil Rehabil* 2018:1-12. doi:
10.1080/09638288.2018.1432704 [published Online First: 2018/02/08]
27. Chan A, Tetzlaff JM, Altman DG, et al. SPIRIT 2013 statement: Defining standard protocol items for
clinical trials. *Annals of Internal Medicine* 2013;158(3):200-07. doi: 10.7326/0003-4819-158-3-
201302050-00583
28. Hoffmann TC, Glasziou PP, Boutron I, et al. Better reporting of interventions: template for
intervention description and replication (TIDieR) checklist and guide. *BMJ (Clinical research ed)*
2014;348:g1687. doi: 10.1136/bmj.g1687 [published Online First: 2014/03/13]
29. Schulz KF, Altman DG, Moher D. CONSORT 2010 statement: updated guidelines for reporting parallel
group randomised trials. *BMJ (Clinical research ed)* 2010;340:c332. doi: 10.1136/bmj.c332
[published Online First: 2010/03/25]
30. Bernet AC, Willens DE, Bauer MS. Effectiveness-implementation hybrid designs: implications for
quality improvement science. *Implementation Science : IS* 2013;8(Suppl 1):S2-S2. doi:
10.1186/1748-5908-8-S1-S2
31. Curran GM, Bauer M, Mittman B, et al. Effectiveness-implementation Hybrid Designs: Combining
Elements of Clinical Effectiveness and Implementation Research to Enhance Public Health
Impact. *Medical care* 2012;50(3):217-26. doi: 10.1097/MLR.0b013e3182408812
32. Lindblad AS, Zingesser P, Sismanyazici-Navaie N. Incentives and barriers to neurological clinical
research participation. *Clinical investigation* 2011;1(12):1663-68. doi: 10.4155/cli.11.153
33. Harris PA, Taylor R, Thielke R, et al. Research Electronic Data Capture (REDCap) - A metadata-driven
methodology and workflow process for providing translational research informatics support.
*Journal of biomedical informatics* 2009;42(2):377-81. doi: 10.1016/j.jbi.2008.08.010
34. Garber CE, Blissmer B, Deschenes MR, et al. American College of Sports Medicine position stand.
Quantity and quality of exercise for developing and maintaining cardiorespiratory,
musculoskeletal, and neuromotor fitness in apparently healthy adults: guidance for prescribing
exercise. *Medicine and science in sports and exercise* 2011;43(7):1334-59. doi:
10.1249/MSS.0b013e318213fefb [published Online First: 2011/06/23]

1 35. Ferguson B. ACSM's Guidelines for Exercise Testing and Prescription 9th Ed. 2014. *The Journal of the*
2 *Canadian Chiropractic Association* 2014;58(3):328-28.
3
4 36. Kietzmann JH, Hermkens K, McCarthy IP, et al. Social media? Get serious! Understanding the
4 functional building blocks of social media. *Business Horizons* 2011;54(3):241-51. doi:
<https://doi.org/10.1016/j.bushor.2011.01.005>
37. Ellis T, Motl RW. Physical activity behavior change in persons with neurologic disorders: overview
and examples from Parkinson disease and multiple sclerosis. *Journal of neurologic physical*
*therapy : JNPT* 2013;37(2):85-90. doi: 10.1097/NPT.0b013e31829157c0 [published Online First:
2013/05/02]
38. Bandura A. Social foundations of thought and action: a social cognitive theory. Englewood Cliffs, NJ:
Prentice Hall 1986.
39. Bandura A. Health promotion by social cognitive means. *Health education & behavior : the official*
*publication of the Society for Public Health Education* 2004;31(2):143-64. doi:
10.1177/1090198104263660 [published Online First: 2004/04/20]
40. Bandura A. Self-efficacy: The exercise of control. New York, NY, US: W H Freeman/Times
Books/Henry Holt & Co. 1997.
41. Maher C, Ferguson M, Vandelanotte C, et al. A Web-Based, Social Networking Physical Activity
Intervention for Insufficiently Active Adults Delivered via Facebook App: Randomized Controlled
Trial. *Journal of medical Internet research* 2015;17(7):e174. doi: 10.2196/jmir.4086 [published
Online First: 2015/07/15]
42. Maher CA, Lewis LK, Ferrar K, et al. Are health behavior change interventions that use online social
networks effective? A systematic review. *Journal of medical Internet research* 2014;16(2):e40.
doi: 10.2196/jmir.2952 [published Online First: 2014/02/20]
43. Befort CA, Donnelly JE, Sullivan DK, et al. Group versus individual phone-based obesity treatment for
rural women. *Eating behaviors* 2010;11(1):11-7. doi: 10.1016/j.eatbeh.2009.08.002 [published
Online First: 2009/12/08]
44. Godin Leisure-Time Exercise Questionnaire. *Medicine& Science in Sports & Exercise* 1997;29(Supplement):36-38. doi: 10.1097/00005768-199706001-00009
45. Rimmer JH, Rauworth A, Wang E, et al. A randomized controlled trial to increase physical activity and
reduce obesity in a predominantly African American group of women with mobility disabilities
and severe obesity. *Preventive Medicine* 2009;48(5):473-79. doi:
<https://doi.org/10.1016/j.ypmed.2009.02.008>
46. Rimmer J, Wang E, Pellegrini C, et al. Telehealth Weight Management Intervention for Adults with
Physical Disabilities A Randomized Controlled Trial 2013.
47. Froehlich-Grobe K, White GW. Promoting physical activity among women with mobility impairments:
a randomized controlled trial to assess a home- and community-based intervention 1 No
commercial party having a direct financial interest in the results of the research supporting this
article has or will confer a benefit upon the author(s) or upon any organization with which the
author(s) is/are associated. *Archives of Physical Medicine and Rehabilitation* 2004;85(4):640-48.
doi: <https://doi.org/10.1016/j.apmr.2003.07.012>
48. Froehlich-Grobe K, Lee J, Aaronson L, et al. Exercise for Everyone: A randomized controlled trial of
Project Workout On Wheels in promoting exercise among wheelchair users. *Archives of physical*
*medicine and rehabilitation* 2014;95(1):20-28. doi: 10.1016/j.apmr.2013.07.006
49. Katherine SB, John PAI, Claire M, et al. Power failure: why small sample size undermines the
reliability of neuroscience. *Nature Reviews Neuroscience* 2013;14(5):365. doi: 10.1038/nrn3475
50. Bailar JC, Iii, Mosteller F. Guidelines for statistical reporting in articles for medical journals:
Amplifications and explanations. *Annals of Internal Medicine* 1988;108(2):266-73. doi:
10.7326/0003-4819-108-2-266

51. Saville DJ. Multiple Comparison Procedures: The Practical Solution. *The American Statistician*
1990;44(2):174-80. doi: 10.2307/2684163
52. Saville DJ. Multiple Comparison Procedures—Cutting the Gordian Knot. *Agronomy Journal*
2015;107(2):730-35. doi: 10.2134/agronj2012.0394
53. P Burnham K, R Anderson D. Multimodel Inference: understanding AIC and BIC in Model
Selection2004.
54. Burnham K AD. Model selection and multimodel inference: A practical information-theoretic
approach. 2 ed: Springer-Verlag 2002.
55. Rimmer JH, Riley B, Wang E, et al. Physical activity participation among persons with disabilities:
barriers and facilitators. *Am J Prev Med* 2004;26(5):419-25. doi: 10.1016/j.amepre.2004.02.002
[published Online First: 2004/05/29]
56. Salisbury SK, Choy NL, Nitz J. Shoulder pain, range of motion, and functional motor skills after acute
tetraplegia. *Arch Phys Med Rehabil* 2003;84(10):1480-5. [published Online First: 2003/10/31]
57. Rimmer JH. The conspicuous absence of people with disabilities in public fitness and recreation
facilities: lack of interest or lack of access? *American journal of health promotion : AJHP*
2005;19(5):327-9, ii. [published Online First: 2005/05/18]
58. Rimmer JH, Padalabalanarayanan S, Malone LA, et al. Fitness facilities still lack accessibility for
people with disabilities. *Disability and Health Journal* 2017;10(2):214-21. doi:
<https://doi.org/10.1016/j.dhjo.2016.12.011>
59. Cook KF, Molton IR, Jensen MP. Fatigue and aging with a disability. *Arch Phys Med Rehabil*
2011;92(7):1126-33. doi: 10.1016/j.apmr.2011.02.017 [published Online First: 2011/06/28]
60. Miriam W, Martin-Diener E, Bauer G, et al. Comparison of trial participants and open access users of
a web-based physical activity intervention regarding adherence, attrition, and repeated
participation. *Journal of medical Internet research* 2010;12(1):e3. doi: 10.2196/jmir.1361
[published Online First: 2010/02/12]
61. Jaarsma EA, Dijkstra PU, Geertzen JH, et al. Barriers to and facilitators of sports participation for
people with physical disabilities: a systematic review. *Scandinavian journal of medicine & science*
*in sports* 2014;24(6):871-81. doi: 10.1111/sms.12218 [published Online First: 2014/04/16]
62. Rimmer JH, Rubin SS, Braddock D. Barriers to exercise in African American women with physical
disabilities. *Arch Phys Med Rehabil* 2000;81(2):182-8. [published Online First: 2000/02/11]
63. Rimmer JH, Silverman K, Braunschweig C, et al. Feasibility of a health promotion intervention for a
group of predominantly African American women with type 2 diabetes. *The Diabetes educator*
2002;28(4):571-80. doi: 10.1177/014572170202800411 [published Online First: 2002/09/13]
64. Rimmer JH, Wang E, Smith D. Barriers associated with exercise and community access for individuals
with stroke. *Journal of rehabilitation research and development* 2008;45(2):315-22. [published
Online First: 2008/06/21]

Protocol Flowchart

224x321mm (300 x 300 DPI)

Template for Intervention Description and Replication

The TIDieR (Template for Intervention Description and Replication) Checklist*:

Information to include when describing an intervention and the location of the information

Item number	Item	Where located **	
		Primary paper (page or appendix number)	Other † (details)
	BRIEF NAME		
1.	Provide the name or a phrase that describes the intervention.	Title Page	_____
	WHY		
2.	Describe any rationale, theory, or goal of the elements essential to the intervention.	1-5	_____
	WHAT		
3.	Materials: Describe any physical or informational materials used in the intervention, including those provided to participants or used in intervention delivery or in training of intervention providers.	8-9	_____
	Provide information on where the materials can be accessed (e.g. online appendix, URL).		
4.	Procedures: Describe each of the procedures, activities, and/or processes used in the intervention, including any enabling or support activities.	10-11	_____
	WHO PROVIDED		
5.	For each category of intervention provider (e.g. psychologist, nursing assistant), describe their expertise, background and any specific training given.	11-12	_____
	HOW		
6.	Describe the modes of delivery (e.g. face-to-face or by some other mechanism, such as internet or telephone) of the intervention and whether it was provided individually or in a group.	10-12	_____
	WHERE		
7.	Describe the type(s) of location(s) where the intervention occurred, including any necessary infrastructure or relevant features.	Title, 18	_____

TIDieR checklist

WHEN and HOW MUCH			
8.	Describe the number of times the intervention was delivered and over what period of time including the number of sessions, their schedule, and their duration, intensity or dose.	9-11	
TAILORING			
9.	If the intervention was planned to be personalised, titrated or adapted, then describe what, why, when, and how.	10	
MODIFICATIONS			
10.†	If the intervention was modified during the course of the study, describe the changes (what, why, when, and how).	N/A	
HOW WELL			
11.	Planned: If intervention adherence or fidelity was assessed, describe how and by whom, and if any strategies were used to maintain or improve fidelity, describe them.	13	
12.‡	Actual: If intervention adherence or fidelity was assessed, describe the extent to which the intervention was delivered as planned.	N/A	

** **Authors** - use N/A if an item is not applicable for the intervention being described. **Reviewers** – use ‘?’ if information about the element is not reported/not sufficiently reported.

† If the information is not provided in the primary paper, give details of where this information is available. This may include locations such as a published protocol or other published papers (provide citation details) or a website (provide the URL).

‡ If completing the TIDieR checklist for a protocol, these items are not relevant to the protocol and cannot be described until the study is complete.

* We strongly recommend using this checklist in conjunction with the TIDieR guide (see *BMJ* 2014;348:g1687) which contains an explanation and elaboration for each item.

* The focus of TIDieR is on reporting details of the intervention elements (and where relevant, comparison elements) of a study. Other elements and methodological features of studies are covered by other reporting statements and checklists and have not been duplicated as part of the TIDieR checklist. When a **randomised trial** is being reported, the TIDieR checklist should be used in conjunction with the CONSORT statement (see www.consort-statement.org) as an extension of **Item 5 of the CONSORT 2010 Statement**. When a **clinical trial protocol** is being reported, the TIDieR checklist should be used in conjunction with the SPIRIT statement as an extension of **Item 11 of the SPIRIT 2013 Statement** (see www.spirit-statement.org). For alternate study designs, TIDieR can be used in conjunction with the appropriate checklist for that study design (see www.equator-network.org).

TIDieR checklist

SPIRIT 2013 Checklist: Recommended items to address in a clinical trial protocol and related documents*

Section/item	Item No	Description	Addressed on page number
Administrative information			
Title	1	Descriptive title identifying the study design, population, interventions, and, if applicable, trial acronym	1
Trial registration	2a	Trial identifier and registry name. If not yet registered, name of intended registry	1
	2b	All items from the World Health Organization Trial Registration Data Set	1
Protocol version	3	Date and version identifier	informed consent
Funding	4	Sources and types of financial, material, and other support	19
Roles and responsibilities	5a	Names, affiliations, and roles of protocol contributors	title page
	5b	Name and contact information for the trial sponsor	19
	5c	Role of study sponsor and funders, if any, in study design; collection, management, analysis, and interpretation of data; writing of the report; and the decision to submit the report for publication, including whether they will have ultimate authority over any of these activities	18
	5d	Composition, roles, and responsibilities of the coordinating centre, steering committee, endpoint adjudication committee, data management team, and other individuals or groups overseeing the trial, if applicable (see Item 21a for data monitoring committee)	n/a

Introduction

Background and rationale	6a	Description of research question and justification for undertaking the trial, including summary of relevant studies (published and unpublished) examining benefits and harms for each intervention	3-5
	6b	Explanation for choice of comparators	4-5
Objectives	7	Specific objectives or hypotheses	5
Trial design	8	Description of trial design including type of trial (eg, parallel group, crossover, factorial, single group), allocation ratio, and framework (eg, superiority, equivalence, noninferiority, exploratory)	5
Methods: Participants, interventions, and outcomes			
Study setting	9	Description of study settings (eg, community clinic, academic hospital) and list of countries where data will be collected. Reference to where list of study sites can be obtained	5-6
Eligibility criteria	10	Inclusion and exclusion criteria for participants. If applicable, eligibility criteria for study centres and individuals who will perform the interventions (eg, surgeons, psychotherapists)	7
Interventions	11a	Interventions for each group with sufficient detail to allow replication, including how and when they will be administered	8-12
	11b	Criteria for discontinuing or modifying allocated interventions for a given trial participant (eg, drug dose change in response to harms, participant request, or improving/worsening disease)	n/a
	11c	Strategies to improve adherence to intervention protocols, and any procedures for monitoring adherence (eg, drug tablet return, laboratory tests)	13
	11d	Relevant concomitant care and interventions that are permitted or prohibited during the trial	7
Outcomes	12	Primary, secondary, and other outcomes, including the specific measurement variable (eg, systolic blood pressure), analysis metric (eg, change from baseline, final value, time to event), method of aggregation (eg, median, proportion), and time point for each outcome. Explanation of the clinical relevance of chosen efficacy and harm outcomes is strongly recommended	13-14; table 3
Participant timeline	13	Time schedule of enrolment, interventions (including any run-ins and washouts), assessments, and visits for participants. A schematic diagram is highly recommended (see Figure)	20

Sample size	14	Estimated number of participants needed to achieve study objectives and how it was determined, including clinical and statistical assumptions supporting any sample size calculations	14-15
Recruitment	15	Strategies for achieving adequate participant enrolment to reach target sample size	5-6
Methods: Assignment of interventions (for controlled trials)
Allocation:
Sequence generation	16a	Method of generating the allocation sequence (eg, computer-generated random numbers), and list of any factors for stratification. To reduce predictability of a random sequence, details of any planned restriction (eg, blocking) should be provided in a separate document that is unavailable to those who enrol participants or assign interventions	12
Allocation concealment mechanism	16b	Mechanism of implementing the allocation sequence (eg, central telephone; sequentially numbered, opaque, sealed envelopes), describing any steps to conceal the sequence until interventions are assigned	11-12
Implementation	16c	Who will generate the allocation sequence, who will enrol participants, and who will assign participants to interventions	8
Blinding (masking)	17a	Who will be blinded after assignment to interventions (eg, trial participants, care providers, outcome assessors, data analysts), and how	12
17b	If blinded, circumstances under which unblinding is permissible, and procedure for revealing a participant's allocated intervention during the trial	n/a
Methods: Data collection, management, and analysis
Data collection methods	18a	Plans for assessment and collection of outcome, baseline, and other trial data, including any related processes to promote data quality (eg, duplicate measurements, training of assessors) and a description of study instruments (eg, questionnaires, laboratory tests) along with their reliability and validity, if known. Reference to where data collection forms can be found, if not in the protocol	13-14; table 3
18b	Plans to promote participant retention and complete follow-up, including list of any outcome data to be collected for participants who discontinue or deviate from intervention protocols	13

Data management	19	Plans for data entry, coding, security, and storage, including any related processes to promote data quality (eg, double data entry; range checks for data values). Reference to where details of data management procedures can be found, if not in the protocol	8
Statistical methods	20a	Statistical methods for analysing primary and secondary outcomes. Reference to where other details of the statistical analysis plan can be found, if not in the protocol	15-16
20b	Methods for any additional analyses (eg, subgroup and adjusted analyses)	15-16
20c	Definition of analysis population relating to protocol non-adherence (eg, as randomised analysis), and any statistical methods to handle missing data (eg, multiple imputation)	15-16
Methods: Monitoring
Data monitoring	21a	Composition of data monitoring committee (DMC); summary of its role and reporting structure; statement of whether it is independent from the sponsor and competing interests; and reference to where further details about its charter can be found, if not in the protocol. Alternatively, an explanation of why a DMC is not needed	16
21b	Description of any interim analyses and stopping guidelines, including who will have access to these interim results and make the final decision to terminate the trial	n/a
Harms	22	Plans for collecting, assessing, reporting, and managing solicited and spontaneously reported adverse events and other unintended effects of trial interventions or trial conduct	15
Auditing	23	Frequency and procedures for auditing trial conduct, if any, and whether the process will be independent from investigators and the sponsor	17
Ethics and dissemination
Research ethics approval	24	Plans for seeking research ethics committee/institutional review board (REC/IRB) approval	15
Protocol amendments	25	Plans for communicating important protocol modifications (eg, changes to eligibility criteria, outcomes, analyses) to relevant parties (eg, investigators, REC/IRBs, trial participants, trial registries, journals, regulators)	15

Consent or assent	26a	Who will obtain informed consent or assent from potential trial participants or authorised surrogates, and how (see Item 32)	14
26b	Additional consent provisions for collection and use of participant data and biological specimens in ancillary studies, if applicable	n/a
Confidentiality	27	How personal information about potential and enrolled participants will be collected, shared, and maintained in order to protect confidentiality before, during, and after the trial	16
Declaration of interests	28	Financial and other competing interests for principal investigators for the overall trial and each study site	19
Access to data	29	Statement of who will have access to the final trial dataset, and disclosure of contractual agreements that limit such access for investigators	16-17
Ancillary and post-trial care	30	Provisions, if any, for ancillary and post-trial care, and for compensation to those who suffer harm from trial participation	informed consent
Dissemination policy	31a	Plans for investigators and sponsor to communicate trial results to participants, healthcare professionals, the public, and other relevant groups (eg, via publication, reporting in results databases, or other data sharing arrangements), including any publication restrictions	16
31b	Authorship eligibility guidelines and any intended use of professional writers	16
31c	Plans, if any, for granting public access to the full protocol, participant-level dataset, and statistical code	n/a
Appendices
Informed consent materials	32	Model consent form and other related documentation given to participants and authorised surrogates	appendix
Biological specimens	33	Plans for collection, laboratory evaluation, and storage of biological specimens for genetic or molecular analysis in the current trial and for future use in ancillary studies, if applicable	n/a

*It is strongly recommended that this checklist be read in conjunction with the SPIRIT 2013 Explanation & Elaboration for important clarification on the items.
 Amendments to the protocol should be tracked and dated. The SPIRIT checklist is copyrighted by the SPIRIT Group under the Creative Commons
 "[Attribution-NonCommercial-NoDerivs 3.0 Unported](https://creativecommons.org/licenses/by-nc-nd/3.0/)" license.
